# Source-resolved atmospheric metal emissions, concentrations, and their deposition fluxes into the East Asian Seas

Shenglan Jiang[1], Yan Zhang[1,2,3*], Guangyuan Yu[1], Zimin Han[1], Junri Zhao[1], Tianle Zhang[4], Mei Zheng[4]

[1]Shanghai Key Laboratory of Atmospheric Particle Pollution and Prevention (LAP3), National Observations and Research Station for Wetland Ecosystems of the Yangtze Estuary, Department of Environmental Science and Engineering, Fudan University, Shanghai 200438, China

[2]Shanghai Institute of Eco Chongming (SIEC), Shanghai 200062, China

[3]MOE laboratory for National Development and Intelligent Governance, Shanghai institute for energy and carbon neutrality strategy, IRDR ICoE on Risk Interconnectivity and Governance on Weather/Climate Extremes Impact and Public Health, Fudan University, Shanghai 200433, China

[4]SKL-ESPC and SEPKL-AERM, College of Environmental Sciences and Engineering, and Centre for Environment and Health, Peking University, Beijing 100871, China

*Correspondence to*: Yan Zhang (yan_zhang@fudan.edu.cn)

**Abstract.** Atmospheric deposition is an important source of marine metallic elements, which have a non-negligible impact on marine ecology. Trace metals from different sources undergo their respective transport processes in the atmosphere, ultimately depositing into the ocean. This study aims to provide gridded data on sea-wide concentrations, deposition fluxes, and soluble deposition fluxes with detailed source categories of metals by the modified Community Multiscale Air Quality (CMAQ) model. A monthly emission inventory of six metals - Fe, Al, V, Ni, Zn, and Cu - from land anthropogenic, ship, and dust sources in East Asia (0-55°N, 85-150°E) in 2017 was developed. Most metals came mainly from land-based sources, contributing over 80%. The annual marine atmospheric deposition fluxes of Fe, Al, V, Ni, Zn, and Cu were 8,827.0, 13,384.3, 99.3, 82.4, 162.7, and 86.5 $\mu g \cdot m^{-2}$, and soluble deposition fluxes were 634.3, 1701.6, 74.3, 46.1, 113.0, and 42.0 $\mu g \cdot m^{-2}$, respectively. Contributions of each source for trace metals varied in emissions, atmospheric concentrations, and depositions. Dust source, as a main contributor of Fe and Al, accounted for a higher proportion of emissions (~90%) than marine deposition fluxes (~20%). However, anthropogenic sources have larger shares of marine deposition flux compared with emissions. The deposition of Zn, Cu, and soluble Fe in East Asian seas was dominated by land anthropogenic sources, while V and Ni were dominated by shipping. The identification of the dominant source of metal deposition offers a foundation for dynamic assessments of the marine ecological effects of atmospheric trace metals. The source-resolved seasonal gridded data makes it possible to calculate soluble metal deposition flux on a source-by-source basis.

## 1 Introduction

Trace metals (iron, cobalt, nickel, copper, zinc, manganese, cadmium, lead, and rare earth elements, among others) have been the focus of marine biogeochemical studies for half a century. They are present in seawater at very low concentrations, typically in the $pmol \cdot L^{-1}$ to $nmol \cdot L^{-1}$ range (Morel and Price, 2003). During the evolution of life, transition metals play a

crucial role in many biochemical functions. It is widely documented that transition trace metals are essential nutrients for
marine biota, such as Fe, Zn, Cu, and Ni (Butler, 1998; De Baar et al., 2018; Whitfield, 2001). Trace metals are involved in
nitrogen and carbon fixation by marine phytoplankton and their mechanism of action is to regulate the expression of biological
enzymes (Bonnet et al., 2008; Browning et al., 2017; Mackey et al., 2015; Morel et al., 1994; Nuester et al., 2012; Rodriguez
and Ho, 2014; Schmidt et al., 2016; Shaked et al., 2006; Sunda, 2012; Tortell et al., 2000; Wuttig et al., 2013a; Wuttig et al.,
2013b). Atmospheric deposition, seafloor hydrothermal upwelling, land-based sediment and riverine inputs, and
remineralization of the oceanic substrate are important sources of marine metals (Longhini et al., 2019; Yang et al., 2019). It
has been shown that the source of atmospheric deposition is important for some elements in seawater, e.g., global atmospheric
deposition of copper is comparable to or even higher than riverine inputs (Little et al., 2014; Takano et al., 2014) and that
atmospheric deposition can carry elements to more remote seas compared to riverine inputs (Yamamoto et al., 2022).
Atmospheric aerosols originate from both natural and anthropogenic sources. Aerosols originating from natural sources
(e.g., dust storms, volcanic eruptions, wildfires) differ significantly in their fluxes, composition, and properties from those
produced by human activities (e.g., industrial emissions, transportation, mining, agriculture) (Baker and Jickells, 2017; Barkley
et al., 2019; Hamilton et al., 2022; Ito et al., 2021; Shi et al., 2023; Zhang et al., 2022). Aerosols from natural sources have
high deposition fluxes and broad deposition ranges, especially for Al and Fe, but generally have low solubility (Baker et al.,
2020; Mahowald et al., 2005; Shi et al., 2015). By contrast, aerosols emitted from anthropogenic sources are often produced
by high-temperature combustion and are characterized by small particle sizes (Bowie et al., 2009; Chen et al., 2012; Li et al.,
2017; Oakes et al., 2012), and contain more soluble metallic elements (Yamamoto et al., 2022; Zhang et al., 2024). To
accurately assess the biogeochemical impact of the atmospheric input, atmospheric particulate species should be determined
for the bioavailable soluble fraction rather than only for the total concentrations or depositions (Birmili et al., 2006; Hsu et al.,
2010). Therefore, emissions from anthropogenic sources, although not as high as those from natural sources, are still of great
concern. Anthropogenic sources can be subdivided into land-based sources and shipping sources. Emissions of ships can be
transported to remote sea areas where land-based aerosols rarely reach. With the development of a booming shipping industry,
their contribution to metal deposition should not be ignored, particularly for V and Ni, which are considered the most abundant
trace metals in heavy ship fuel oils (HFO) (Celo et al., 2015; Corbin et al., 2018).
The spatial distribution of metal emissions from ship and anthropogenic sources contrasts with that of dust (Mahowald et
al., 2018). Dust has long been considered an important source of Fe to the surface ocean, particularly in remote areas away
from continental margins (Jickells et al., 2005). However, Matsui et al. (2018) suggested that anthropogenic Fe may dominate
the total deposition flux of soluble Fe and its variability over southern oceans (30-90°S) by incorporating recent measurements
of anthropogenic magnetite into a global aerosol model, which increased the estimated total deposition flux of soluble Fe to
southern oceans by 52%. Pinedo-González et al. (2020) determined from iron-stable isotopes that anthropogenic Fe contributed
21-59% of soluble Fe measured in the North Pacific Ocean. The Northwest Pacific is located directly downwind of the
industrially active East Asian region with significant and increasing metal emissions and is influenced by westerly winds
transporting Asian dust (often mixed with anthropogenic aerosol and gases) (Hamilton et al., 2023). Identifying the dominant
sources of metal deposition in the ocean is important for estimating soluble metal deposition, especially in the East Asian seas
with significant contributions from both dust and anthropogenic metal emissions.
Current studies on metal emission inventories mainly focused on land-based emission sources (Bai et al., 2021; Tian et al.,
2015; Wang et al., 2016). The inventories including high-resolution ship sources only covered a limited number of metals such
as V and Ni (Zhai et al., 2023; Zhao et al., 2021a), yet the contribution of shipping to other metals should also be considered.
Previous studies about the concentration and deposition flux of metals were done by site observations and source
apportionment by statistical methodologies (Fu et al., 2023; Okubo et al., 2013; Pan and Wang, 2015; Pan et al., 2021; Tao et
al., 2016; Tao et al., 2017; Wei et al., 2014; Zhang et al., 2024). Due to limitations in the location of the observation sites,
these studies were unable to provide data over a wide area of the ocean and there was uncertainty in confirming the source
based on statistical methods. Current model-based simulations of gridded concentrations, deposition fluxes, and distinguishing
between sources were mainly focused on Fe (Matsui et al., 2018; Yamamoto et al., 2022). The broader regional scale study by
air quality model was few maybe due to the shortage of emission inventories of trace elements. The emission inventories,
including metals with marine ecological effects and metals representative of dust and ship sources, need to be developed.
Additionally, the atmospheric transport processes of these metals and their deposition fluxes to the ocean remain to be studied.
In this study, we established an emission inventory of six metal elements (Fe, Al, V, Ni, Zn, Cu) from three major emission
sources, namely, land anthropogenic, ship, and dust sources, in the East Asian region (0-55°N, 85-150°E) in 2017. The aerosol
module in the Community Multiscale Air Quality (CMAQ) model was modified to simulate the concentration, dry and wet
deposition fluxes of the metallic elements, and calculated the soluble metal deposition fluxes. In addition, we quantified the
contribution of each source to the emissions and concentrations of metal elements in East Asia and analyzed the sources of
deposited metals in different sea areas.

## 2 Materials and Methods

### 2.1 Description of the Modelling System

The CMAQ (E.P.A, 2020) is a widely used air quality model that encompasses a wide range of complex atmospheric
physicochemical processes. This study modeled metal concentrations and dry and wet deposition fluxes using the CMAQ
version 5.4. The multi-pollutant code in the aerosol module and the in-line dust emission module of CMAQ v5.4 were modified
to add metallic elements as modeling variables. In the revised version of the CMAQ model, it was assumed that these 6 metallic
elements were considered inert chemical constituents in aerosols, which can participate in atmospheric physical processes such
as diffusion, advection, and deposition, but do not participate in any atmospheric chemical reactions. Specific modifications
are described in the Supporting Information (Text S1).
The CMAQ model configuration utilized AERO7 for the aerosol module (Xu et al., 2018) and CB6r5 for the gas-phase
mechanism (Amedro et al., 2020), including detailed halogen chemical components (Sarwar et al., 2019) and DMS (Lana et
al., 2011; Zhao et al., 2021b). M3Dry scheme was used to calculate dry deposition (Pleim and Ran, 2011), and the aerosol dry
deposition model was upgraded in version 5.4, showing better comparison with size-resolved observations (Pleim et al., 2022);
AQCHEM cloud chemistry was used to calculate wet deposition (Fahey et al., 2017). Initial and boundary conditions for the
simulation domain were established based on seasonal average hemispheric CMAQ output from the CMAS data repository
(E.P.A, 2019). Meteorological fields were generated using the Weather Research and Forecasting (WRF) model version 4.1.1,
with initial and boundary conditions sourced from the 6-hour temporal resolution National Centers for Environmental
Prediction (NCEP) Final Operational Global Analysis dataset. The physics schemes are listed in the Supporting Information
(Text S2).
In this study, three scenarios were carried out to investigate the whole process from emission to atmospheric concentration
to deposition in the sea and the effects of different emission sources on atmospheric concentration and deposition fluxes of
metals. One scenario included three emission sources: land anthropogenic, ship, and dust sources. Another scenario included
only land anthropogenic and dust sources. The other scenario included only land anthropogenic and ship sources.  The
contributions of ship and dust sources to metal concentrations and deposition fluxes were extracted based on the zero-out
method, i.e., two runs with and without ship or dust emissions. And the impact of land anthropogenic sources was further
calculated. Each simulation was conducted for January, April, July, and October of 2017 with a 5-day spin-up period to
calculate the atmospheric concentrations and deposition fluxes of metals, representing winter, spring, summer, and autumn,
respectively. The simulation domain covers East Asia and most of the East Asian Seas, as shown in Fig. S1, discretized with
a horizontal grid resolution of 36 km and 27 vertical layers between the surface and 100 hPa, and the surface layer thickness
was ~40 m.
**2.2 Methodology of Metal Emission Inventory**
In this study, metal emission sources were categorized into land anthropogenic, ship, and dust sources. The general
methodology for calculating monthly land anthropogenic emissions of metals was to multiply each source of PM emissions
by the fraction of the metal content in PM. Monthly emissions data for 2017 for each source category of PM was provided by
the Emissions Database for Global Atmospheric Research (EDGAR) emission inventories (Crippa et al., 2020) (global, 0.1 ×
0.1° resolution), and corresponding source-specific speciation profiles were created based on the SPECIATE v5.1 database
(Bray et al., 2019; Simon et al., 2010) The same approach was used in previous metal emission inventories (Gargava et al.,
2014; Kajino et al., 2020; Reff et al., 2009; Xuan, 2005; Ying et al., 2018).
The monthly emission inventory of metals from ship sources was established by a bottom-up approach based on real-time
data from the Automatic Identification of Ships (AIS) database for the year 2017 (Yuan et al., 2023; Zhao et al., 2020).
Parameters such as power-based emission factors (in g·kWh$^{-1}$) are listed in the Supporting Information (Table S1 and S2) and
the low load adjustment multipliers can be found in the previous studies (Chen et al., 2017; Fan et al., 2016). More information
on the emission inventories can be found in the Supporting Information (Text S3).
The monthly dust emissions of trace metals in 2017 were generated from in-line modules developed by Foroutan et al. (2017)
during the CMAQ run. We modified the in-line windblown dust module to incorporate metal species, facilitating its concurrent
operation with the MODIS land cover data. For the dust speciation factor, we adjusted the fine and coarse mode mass fractions
of metal species based on a comprehensive literature review. The detailed findings of the literature review, along with the
ultimately modified values, are presented in Table S3.
**2.3 Calculation of soluble metal deposition fluxes**
In this study, the soluble fraction of the metal deposition flux was roughly calculated by multiplying the deposition flux
obtained from the CMAQ simulation by the solubility of the metal, which has also been used in previous studies (Liu et al.,
2022; Zhang et al., 2024). The solubility of metals is closely related to the source (Chester et al., 1993). Kurisu et al. (2021)
used the stable Fe isotope source apportionment method to analyze dust Fe and anthropogenic Fe concentrations in total and
soluble Fe samples. The results showed that the solubility of dust Fe in the Northwest Pacific Ocean ranged from 0.9 ~ 1.3%
(dust-contributed soluble Fe divided by dust-contributed total Fe) and 11% for solubility of anthropogenic Fe (anthropogenic-
contributed soluble Fe divided by anthropogenic-contributed total Fe). However, a large number of observations reported
samples with iron solubility in the marine atmosphere exceeding 10% (Gao et al., 2013; Shi et al., 2013; Sholkovitz et al.,
2012), which illustrates the fact that a rough classification of sources into dust and anthropogenic sources is not sufficiently
plausible and that sources of emissions of highly soluble metals such as shipping, for example, need to be considered as well
(Ito, 2015). This study distinguished the contribution of different sources to the deposition flux of metals, providing the
possibilities for considering the distinct solubilities of metals from various sources. Given that current studies primarily focused
on Fe, obtaining solubility data for other metals from different sources proved challenging. The solubility adopted in this study
is shown in Table S4, which differentiated between fine and coarse modes and three emission sources for Fe, and only two
modes for the other metals.
**3 Results and Discussion**
**3.1 Emission Inventory**
**3.1.1 Contributions of Various Sectors**
We used monthly emission inventories from land anthropogenic and ship sources and modelled monthly dust emissions for
2017 to calculate metal emissions for the entire year. The relative contribution of the three sources to metal emissions and the
seasonal variation characteristics were assessed, and then emissions from land anthropogenic sources were further specified.
As shown in Fig.1, for the fine mode of six metals, emissions originating from land anthropogenic sources were much more
significant than those from ship or dust sources, with relative contributions largely exceeding 59% and peaking at 95.2%. The
emissions from ship sources were not large overall, but the relative contribution to fine mode V and Ni could reach 21.4% and
13.4%, which is similar to the results of previous studies on ship emissions (Yuan et al., 2023; Zhao et al., 2021a). Dust
substantially released Fe and Al in coarse mode (accounting for 79.6% and 87.4% of the coarse mode emissions, respectively),

while showing rather low contribution to other metals, which was related to the content of metallic elements in soil minerals. Land anthropogenic sources showed higher emissions of Fe and Al elements, amounting to 208.1 and 242.2 kt·year$^{-1}$ respectively. In contrast, V and Ni showed a lesser degree of impact from land anthropogenic activities, with values of 8.2 and 9.4 kt·year$^{-1}$. V showed the highest fine and coarse mode ratio of 4.6, while Cu showed a ratio of 1.1.

The monthly emission statistics of the three sources are detailed in Table S5-S9. According to Table S5, the overall quantity of metals emitted by ships was predominantly higher in summertime (July and August), followed by wintertime (November and December), while it was relatively lower in September. This is related to the activities of ships, which are more active in the summer months and have higher emissions, a trend that has been reported in previous studies (Chen et al., 2018; Zhai et al., 2023). In terms of land anthropogenic sources (Tables S6-7), the emissions of all metallic elements in the fine mode were greater in winter (December and January) due to elevated heating demand, a seasonal feature consistent with previous studies (Luo et al., 2022; Zhao et al., 2021c). The emissions of Fe, Al, and Ni in the coarse mode showed the same seasonal characteristics, while the highest emissions of V, Cu, and Zn occurred in April and October. Overall, the monthly variation of metal emissions from land anthropogenic sources was not as significant as that from ship sources, suggesting that metals could be emitted from stable sources such as industrial combustion (Zhang et al., 2018). Dust emissions were mainly concentrated in April, accounting for about 45% of the total annual emissions. In consideration of the significant seasonal variation, we counted the contribution of metals from the three emission sources in spring, as shown in Fig.S2. Dust sources were identified as the primary contributor to the coarse mode emissions of Fe and Al, accounting for a higher proportion of spring emissions than of annual emissions, 90.0% and 94.2% respectively. For the fine mode springtime emissions of these two metals, dust sources accounted for 51.9% and 61.8%, respectively, and were also the most significant source of emissions. There were also relatively high emissions in July and May, with the remaining months being insignificant. This is related to the fact that dust events in East Asia occur mainly in spring (Gui et al., 2022; Hsu et al., 2010; Kang and Wang, 2005; Kang et al., 2016) and studies have also reported dust events in summer (Chen et al., 2014) and autumn (Zhang et al., 2015) in certain years.

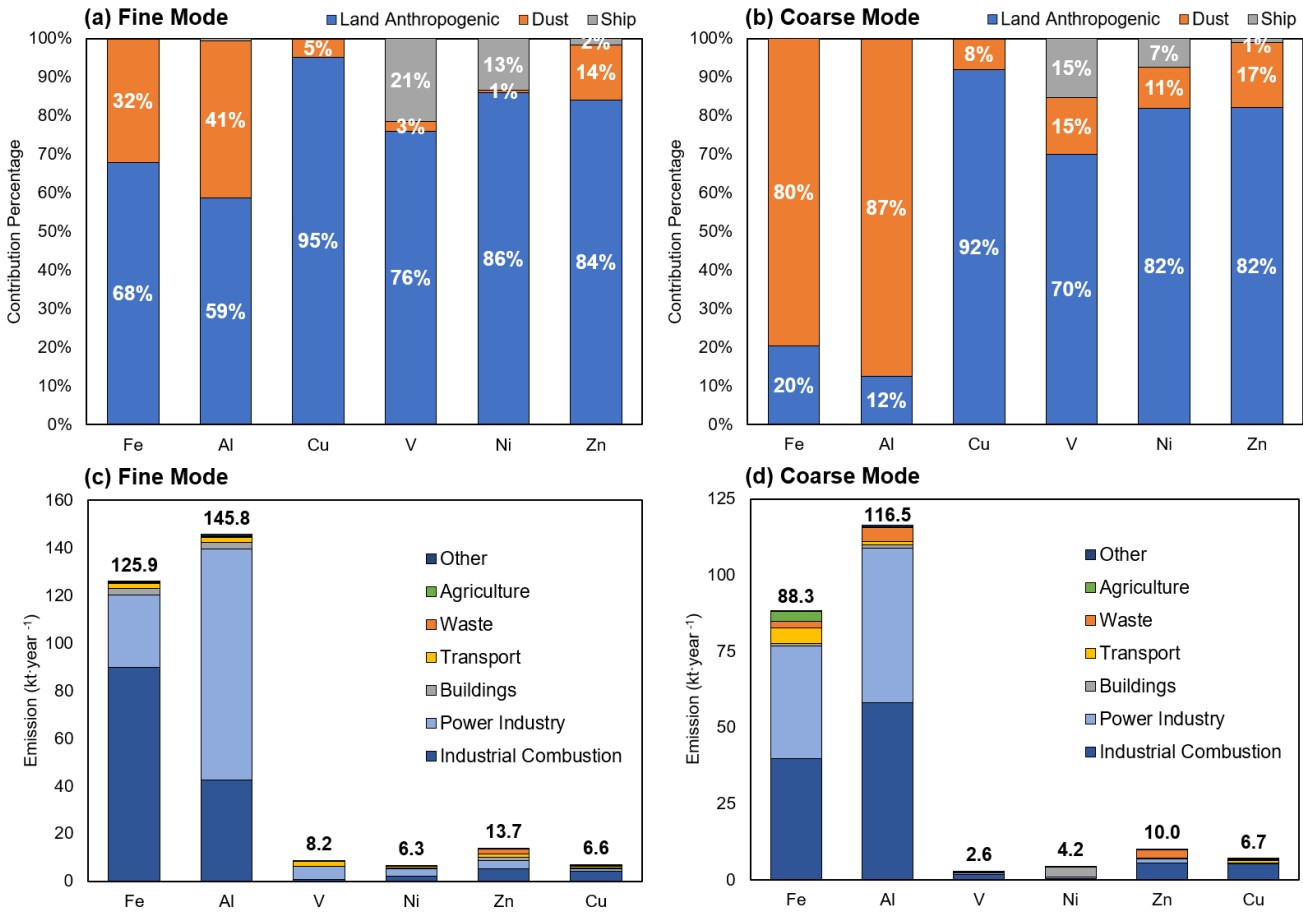

184

Figure 1: Relative contributions of land anthropogenic, ship, and dust sources to fine mode (a), coarse mode (b) emissions of the six metals (Fe, Al, V, Ni, Zn, Cu); stacked histograms of the absolute contributions of the seven emission sectors of land anthropogenic sources to fine mode (c), coarse mode (d), with the numbers representing the total emissions from all anthropogenic emission sectors.

The predominant sources of emissions, specifically land anthropogenic sources, were further classified into seven categories according to EDGAR, namely Industrial Combustion, Power Industry, Buildings, Transport, Waste, Agriculture, and Other (Figs.1c and 1d). For all six metals, both the Power industry and Industrial Combustion sources emerged as the prominent contributors, collectively accounting for more than 50% of the total land anthropogenic emissions. The emissions of Fe originating from industrial combustion were the largest, amounting to 129.5 kt·year$^{-1}$, with the fine mode accounting for 69.4%. The emissions of Al from the power industry were significant, amounting to 148.0 kt·year$^{-1}$, with the fine mode accounting for 65.7%. In addition, the waste sector made a substantial contribution to Zn with 5.0 kt·year$^{-1}$, which was comparable to the 4.6 kt·year$^{-1}$ contributed by the power industry. And the metals emitted from the waste sector were mainly in coarse mode, the proportion of coarse mode was more than 80%, except for Cu (24.8%) and Zn (55.9%).

Several studies on metal emission inventories (refer to Table S10) are accessible for conducting comparative analyses. In the context of land anthropogenic sources, the emissions of Ni were reported as 3,395.5 tons, Zn as 22,319.6 tons, and Cu as

9,547.6 tons in China in 2012 (Tian et al., 2015). Additionally, V emissions, inclusive of land anthropogenic and dust emissions,
were documented as 11,505.04 tons in China in 2017 (Bai et al., 2021). In this study, the corresponding values (ensuring
consistency of emission sources and areas) were 5,494.5 tons for Ni, 13,407.2 tons for Zn, 6,578.9 tons for Cu, and 11,093.7
tons for V in 2017. In terms of subdivided emission sectors, solid waste contributions were 0.3, 43.5, 1,790.7, and 382.4
tons·year$^{-1}$ for V, Ni, Zn, and Cu, respectively (Bai et al., 2021; Wang et al., 2017b), and 0.6, 27.9, 2,194.0, and 185.6 tons·year$^{-1}$
in this study. The Iron and Steel sector emitted 79.6 and 105.0 tons·year$^{-1}$ of V and Ni (Bai et al., 2021; Wang et al., 2016),
compared to 109.2 and 196.0 tons·year$^{-1}$ in this study. The ship emissions of V and Ni in East Asia in 2015 reported by Zhao
et al. (2021a) were 1,329.8 and 580.4 tons/year, while in this study, they were 1,802.6 and 854.8 tons·year$^{-1}$, with an acceptable
range of differences. Considering the different base years of the inventories and the different types of anthropogenic sources
covered, the results of this study were consistent with previous studies overall.
**3.1.2 Spatial distribution of metal emissions**
The spatial distributions of metal emissions were presented in Figs.2a-2f. For the entire simulation area, the emissions of
Fe, Al, V, Ni, Zn, and Cu from all sources were 1,021.5, 1,940.4, 11.7, 11.5, 27.2, and 14.0 kt in 2017, respectively. In the
context of the modelled land area, China was found to release substantial amounts of Fe, Al, V, Ni, Zn, and Cu, totaling
810,869.5, 157,099.8, 7,994.9, 7,639.7, 18,838.1, and 10,225.6 tons·year$^{-1}$, respectively. Beyond China, significant emissions
were found in the coastal cities of Japan and South Korea, as well as in Southeast Asian regions. Specifically, Japan and South
Korea contributed 6,239.5, 4,545.3, 190.7, 197.3, 538.8, and 424.6 tons·year$^{-1}$ to the six metals, respectively. The emissions
from India were 37,717.2, 54,059.0, 1,059.3, 2,028.7, 3,057.3, and 1,754.0 tons·year$^{-1}$, respectively. Meanwhile, the emissions
from Southeast Asia were 6,315.9, 10,249.2, 258.0, 607.8, 747.0, and 407.0 tons·year$^{-1}$. Significantly emissions in the North
China Plain, the Yangtze River Delta, the Pearl River Delta, and Central China can be attributed to dense human activity levels
in these regions, as reported by previous study (Bai et al., 2021). Notably, the dust source regions of East Asia, namely the
Taklamakan Desert and the Mongolian Plateau, showed remarkable emissions of Fe and Al, surpassing those of densely
populated and economically developed regions by an order of magnitude or more.

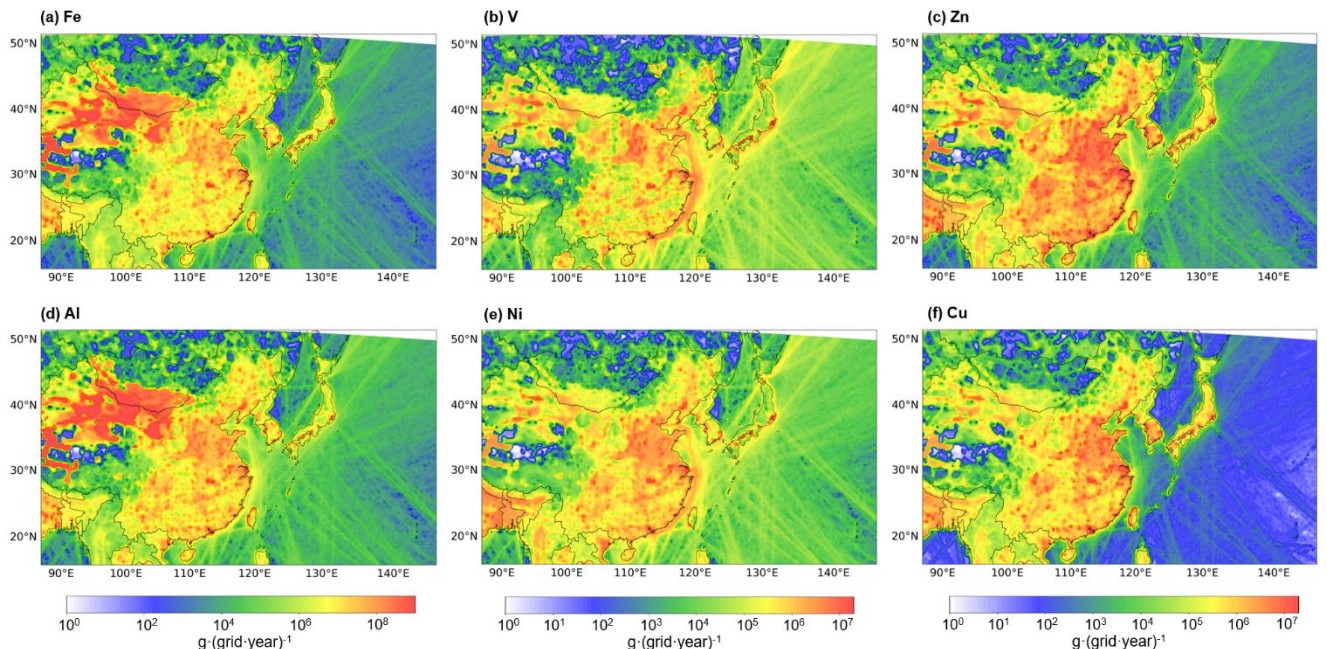

Figure 2: Girded metal emissions from all sources for the year 2017 (36 km ×36 km resolution; units, grams per year per grid cell, including land anthropogenic, ship, and dust sources). Fe (a), V (b), Zn (c), Al (d), Ni (e), Cu (f). See Table S5-9 for detailed emission data information.

Within the marine domain, the emission trajectories of V and Ni were more substantial than the rest of the metals, as Fig.2 illustrated. In the coastal waters of eastern China, ship activities are dynamic, creating a linear high-emission zone in areas with dense shipping routes, and the emissions of V and Ni brought by ships were comparable to the contribution of land anthropogenic sources. By contrast, the emissions of the remaining four metals in the marine area were notably lower than those in the land area. Furthermore, ships represent in-site sources of marine pollution, their emission footprint covers the vast expanse of the Pacific Ocean, highlighting the importance of considering ship sources in emission inventories.

## 3.2 Contributions of different sources to marine atmospheric metal concentrations and deposition fluxes

### 3.2.1 Contributions of different sources to marine atmospheric metal concentrations

Based on the emission inventory of metallic elements established in Sect.3.1, the concentrations of metals in the sea areas and the contributions of different sources were simulated by the CMAQ model. Overall, the seasonal mean metallic concentrations in sea areas were 34.9, 51.3, 1.0, 0.6, 1.0, and 0.5 ng·m$^{-3}$ for Fe, Al, V, Ni, Zn, and Cu, respectively. It is worth noting that we chose January, April, July, and October to represent each of the four seasons, and since most of the spring dust events in East Asia occur in April, this estimate would result in a slight overestimation of the contribution of dust sources. Concentrations in the Bohai Sea (BS) and the Yellow Sea (YS) were significantly higher than those in the other seas, about 5-20 times higher than the sea-wide average (Fig.3). The BS demonstrated the highest concentrations of five metallic elements,

Fe, Al, Ni, Zn, and Cu, at 225.3, 249.7, 10.4, 18.7, and 9.6 ng·m$^{-3}$, respectively. The YS showed a notably higher concentration
of V (15.34 ng·m$^{-3}$), which was attributed to dense ship activities in the marginal sea of China. Dust sources predominantly
influenced the concentrations of Fe and Al, accounting for 17.9% and 28.5%, on the sea-wide average, and their contributions
to the remaining four elements were far less than those from land anthropogenic or ship sources. Asian dust storms occur
annually in late winter and spring in the main dust regions of the Gobi Desert, Taklamakan Desert, and Loess Plateau (Hsu et
al., 2010). Therefore, dust sources played a more significant role in April, contributing 39.2% of the Fe and 51.3% of the Al
concentrations in the sea area covered by the study. In the East China Sea (ECS), these values could reach 48.3% and 67.8%,
respectively (as presented in Fig.S3). Ship sources mainly contributed to the concentrations of V and Ni in the sea area, with
average contribution shares of 56.4% and 37.8%, and can reach 65.7% and 49.3% in the ECS, respectively. Land anthropogenic
sources were the most important contributors to the sea level concentrations of most of the metal elements, excluding V, with
an average contribution of 42.7%. Notably, for Cu, which is not a major metal element emitted from ships and whose content
in dust particles is relatively small, the contribution from anthropogenic sources was as high as 97.6%. The concentration of
Fe was 201.1 ng·m$^{-3}$ in the YS and 92.17 ng·m$^{-3}$ in the ECS, and the contribution of land anthropogenic sources to the Fe
concentration was 71.6% in the ECS, similar to the values reported by previous study (Zhang et al., 2024). The available long-
term and near real-time concentration monitoring data of V and Ni in the fine mode at the Pudong site (in Shanghai, China)
obtained by Zou et al. (2020) were used to further validate the simulation of the model. As presented in Fig.S4, the simulated
concentrations of V and Ni were in good agreement with the monitoring data, with respective normalized mean fractional bias
(NMFB) and normalized mean fractional error (NMFE) of -0.31 and 0.37 for V and -0.38 and 0.40 for Ni. Additionally, Table
S11 presents a comparison between the metallic element concentrations in the East Asian land region and the simulation results
derived from this study.

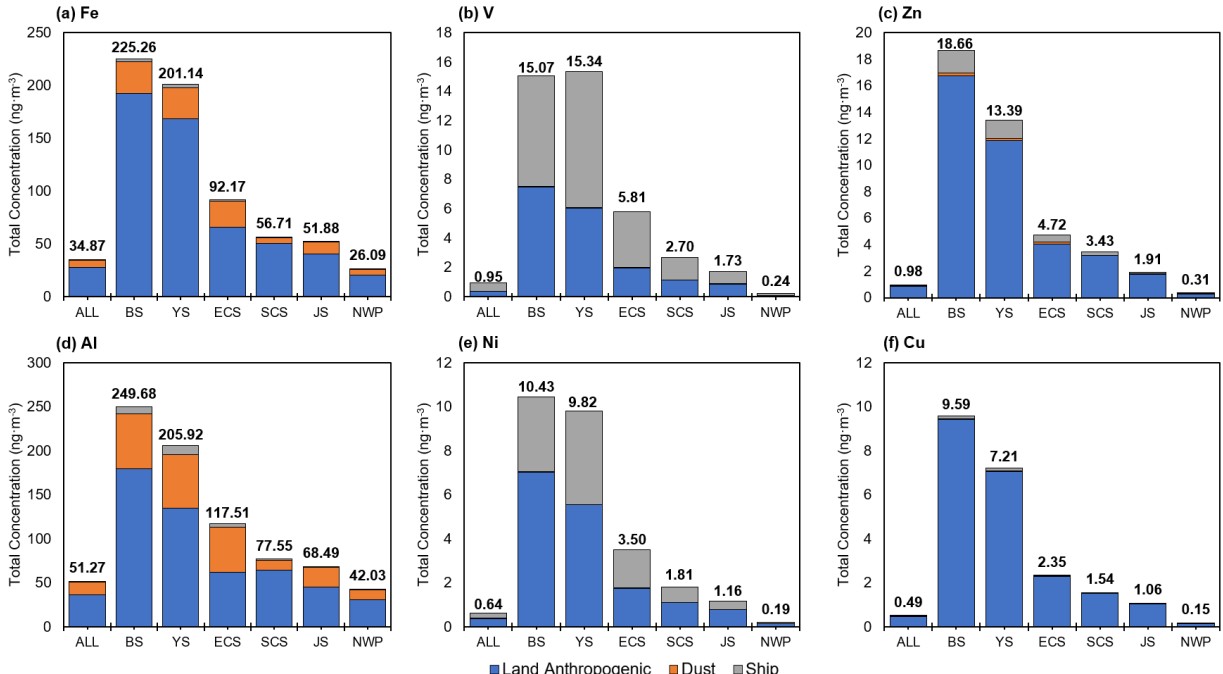


**Figure 3: Contributions of seasonal mean concentrations of metallic elements in different sea areas from land anthropogenic, ship, and dust sources, Fe (a), V (b), Zn (c), Al (d), Ni (e), Cu (f) (units: ng·m$^{-3}$), with the numbers at the top of the stacked bar charts representing the total seasonal mean concentrations from the three major sources.**

Land anthropogenic, ship, and dust sources presented discernible differences in both absolute and relative contributions of metal elements across diverse sea areas. Moreover, metallic element concentrations originating from these three sources showed distinct spatial distributions. As illustrated in Fig.S5, all six metal concentrations attributed to land anthropogenic sources were notably higher in coastal areas, particularly in the proximity of China and Korea. Because land anthropogenic metals are mainly transported by diffusion and advection rather than strong weather processes, which is different from dust sources. Studies have shown that the outbreak of Asian dust storms is often associated with the Mongolian cyclones during spring (Gui et al., 2022). This atmospheric phenomenon results in the transport of metal particles from natural dust sources to more open sea areas, rather than being confined to coastal areas, and these metal particles show a spatial distribution pattern following the trailing flow of the cyclone. As shown in Fig.S3, dust sources contributed 40.8% and 50.3% of the atmospheric concentrations of Fe and Al in the NWP in spring, respectively. Due to the higher contents of Fe and Al elements in soil, concentrations of Fe and Al resulting from dust were 2-3 orders of magnitude higher than those of the other four metals. Metal concentrations from ship sources were predominantly distributed around busy shipping routes, with higher concentrations within the 200 nautical miles (nm) range of East Asian countries. However, high concentration values were noted at a certain distance from the coastline, distinct from the concentration distribution of land anthropogenic sources.

### 3.2.2 Contributions of different sources to marine atmospheric metal deposition fluxes

The influence of the three emission sources on metal deposition fluxes and concentrations across the sea areas displayed distinctive characteristics. As depicted in Fig.3, the concentrations of six metal elements over the BS and the YS markedly surpassed those recorded in other seas, and were even 6-60 times higher than those over the open Northwest Pacific Ocean (NWP). However, the deposition fluxes of metal elements over proximate coastal areas, including the BS, the YS, the ECS, the South China Sea (SCS), and the Sea of Japan (JS), showed relatively insignificant differences, although the BS and the YS still displayed the highest fluxes (Fig.4). It can be seen that the spatial distribution of metal deposition in the seas was broader than that of metal concentrations (Fig.S6). Similar to Sect. 3.2.1, deposition fluxes from land anthropogenic and ship sources during representative months of the four seasons were used to estimate the deposition fluxes for the corresponding seasons to calculate the estimated annual values, an estimation method that has been used in previous studies (Lin et al., 2010; Zhang et al., 2010). Given the considerable seasonal variability of dust sources, we employed a conversion factor to estimate the seasonal values via monthly deposition fluxes, which was derived from the ratio of the total seasonal emissions from dust sources to the emissions in a representative month. For example, if the dust emissions in spring (March-April-May) are 1.27 times the dust emissions in April, the spring deposition flux from dust sources is calculated as the deposition flux from the April dust contribution multiplied by 1.27. Table S12 presents the comparison of the stimulated deposition fluxes of the metals in this study with existing observation-based studies on metal deposition fluxes. Given that the existing studies focused more on the land area, this study employed land deposition flux data for comparison. The deposition fluxes of the six metals were within the range of the existing studies, validating our results.

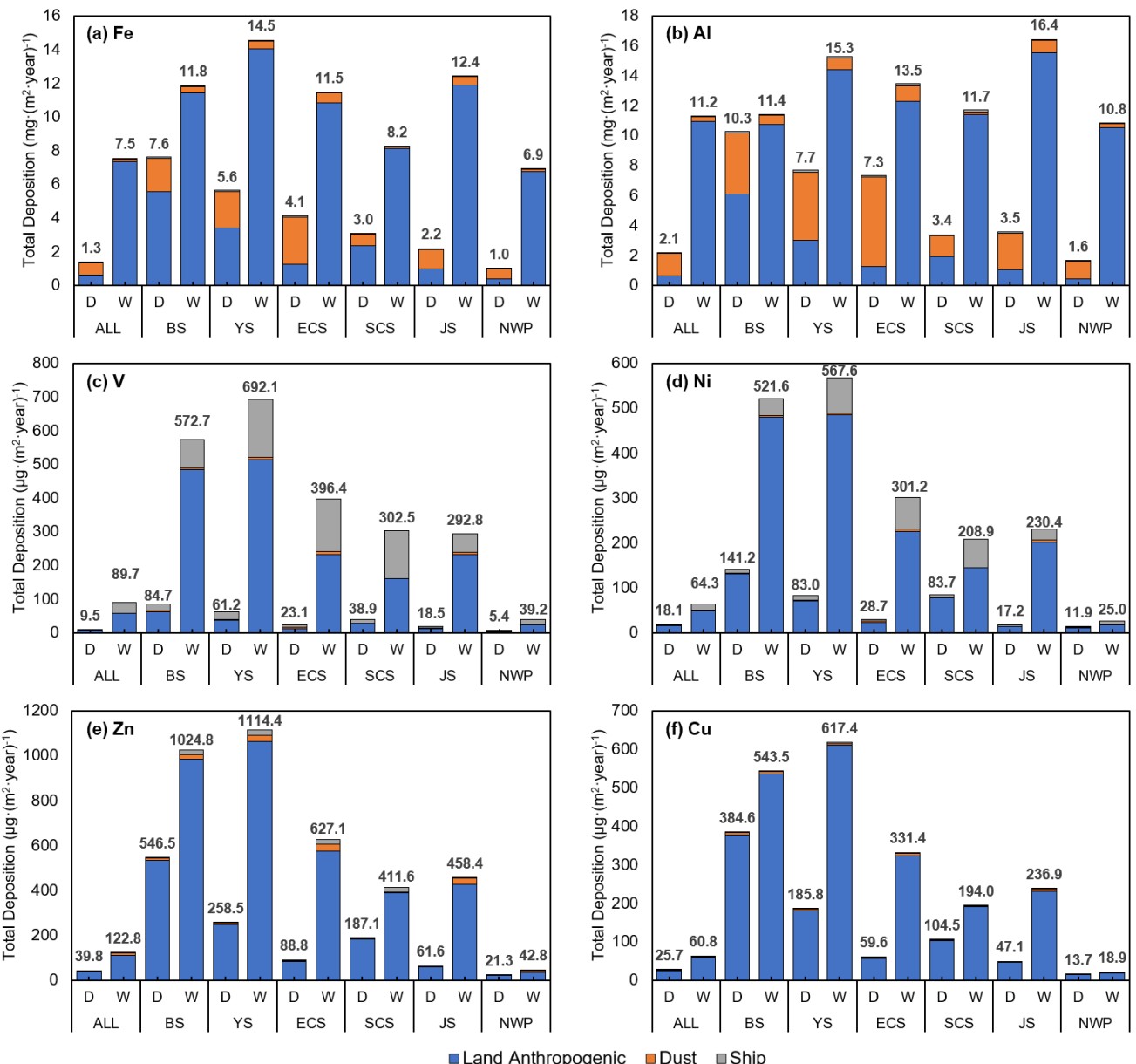

**Figure 4: Contributions of land anthropogenic, ship, and dust sources to the estimated annual dry and wet deposition fluxes of**
**metallic elements (represented by D and W, respectively, in the figures) in different marine areas, Fe (a), Al (b), V (c), Ni (d), Zn (e),**
**Cu (f) (units: mg·m$^{-2}$·year$^{-1}$ for Fe and Al, μg·m$^{-2}$·year$^{-1}$ for V, Ni, Zn, Cu), and the numbers above the stacked bars represent the**
**total annual dry or wet deposition fluxes from the three major sources.**
Particulate elements are removed from the atmosphere through dry and wet deposition processes, and wet deposition is
generally more important than dry deposition in marine areas (Mahowald et al., 2005). At the marine scale, wet deposition
fluxes were greater than dry deposition for all six metal elements, which is in line with previous findings (Connan et al., 2013;
Gao et al., 2013; Zhang et al., 2024). The dry and wet deposition ratios (i.e., dry deposition flux/wet deposition flux) of Fe, Al,

V, Ni, Zn, and Cu were 0.18, 0.19, 0.11, 0.28, 0.32, and 0.42 across the entire study sea area, respectively. Dry deposition flux is a function of atmospheric concentration and particle dry deposition velocity. Wet deposition removes airborne particulate elements via precipitation scavenging, which includes in-cloud and below-cloud scavenging (Cheng et al., 2021). The size distribution of metals in atmospheric aerosols is a key factor influencing the differences between wet and dry deposition flux. Sakata and Asakura (2011) indicated that metals associated with coarse particles (> 2.5 μm in diameter) have shorter atmospheric lifetimes due to gravitational settling and inertial deposition, which easily govern dry deposition. Fine particulate matter, on the other hand, is more likely to serve as condensation nuclei for wet deposition. Dust sources, typically characterized by large particle sizes, are consequently more readily removed from the atmosphere through dry deposition during atmospheric transport. The fine mode proportion of the six metals from both land anthropogenic and ship sources were, in descending order, V (82%), Ni (62%), Fe (60%), Zn (60%), Al (59%), and Cu (51%), and anthropogenic sources contributed more than dust sources. As a result, these sources contributed predominantly to metal deposition in the sea through wet deposition processes. The difference in particle size and behavior highlights the complex interplay between source-specific attributes and deposition mechanisms, influencing the fate of metals in the atmosphere and their subsequent deposition in the ocean.

The spatial distribution of annual deposition fluxes of six metals in the sea was illustrated in Fig.S6. Over the whole sea area, the estimated annual deposition fluxes of Fe, Al, V, Ni, Zn, and Cu were 8,827.0, 13,384.3, 99.3, 82.4, 162.7, and 86.5 μg·m$^{-2}$·year$^{-1}$, respectively, in which the highest values of deposition fluxes reached 246.5, 246.2, 7.4, 3.3, 16.9, and 11.0 mg·m$^{-2}$·year$^{-1}$. The deposition of Fe and Al in the sea showed a wider spatial extent compared to the other four metals, particularly in the NWP. Combined with Fig.S5, it can be hypothesized that this phenomenon was caused by dust sources, as metallic particulate matter was transported and deposited into the more open ocean along with intense weather processes like cyclones and cold fronts (Li and Chen, 2023). During the spring season, when dusty weather is frequent, the contribution of dust sources to the deposition fluxes of Fe and Al in the whole sea area reached 50.9% and 60.5%, respectively, and the contribution to the NWP can also reach 49.2% and 57.3%, respectively. The deposition of V, Ni, Zn, and Cu, was primarily distributed in offshore waters, such as the BS, the YS, and the JS, as well as within the 100 nm range in eastern China. The deposition fluxes of V were high in the 200 nm range in eastern China, which is related to the ship activities, as reported by previous study (Zhao et al., 2021a).

### 3.2.3 Estimation of Deposition Flux of Soluble Metals in Maritime Areas

Utilizing the calculation methods in Sect.2.3, the detailed results of these calculations for soluble metal deposition fluxes to the ocean within the study area were provided in Table 1. In this context, the soluble Fe deposition flux was calculated separately for each of the three sources and then summed to obtain the total soluble deposition flux. Land anthropogenic, ship, and dust sources contributed 600.0, 10.6, and 1.7 μg·m$^{-2}$·year$^{-1}$ of soluble Fe in the fine mode and 10.9, 0, and 12.0 μg·m$^{-2}$·year$^{-1}$ of soluble Fe in the coarse mode, respectively. Based on this method, the solubility of Fe (soluble Fe from all three

sources divided by total Fe deposition flux) obtained in this study ranged from 4% to 17%, which is comparable to the results
of previous studies (Alexander et al., 2009; Kurisu et al., 2021; Shao et al., 2019).
**Table 1: Marine deposition fluxes of soluble metals in fine and coarse particulate forms (Units: $\mu g \cdot m^{-2} \cdot year^{-1}$)**

|  | Cu | Fe* | Zn | V | Ni | Al |
|---|---|---|---|---|---|---|
| Fine mode | 26.1 | 611.4 | 80.2 | 72.4 | 41.8 | 1608.9 |
| Coarse mode | 15.8 | 22.9 | 32.7 | 1.9 | 4.3 | 92.7 |

*The soluble iron deposition flux was calculated separately for each of the three sources and then summed to obtain the total soluble
deposition flux
Figure 5 illustrated the spatial distribution of fine and coarse mode soluble Fe deposition over different sea areas, and Fig.S7
showed the absolute and relative contributions of the three sources to soluble Fe deposition over these areas. The spatial
distribution displayed marked differences for different particle sizes. The deposition fluxes of fine-mode soluble Fe were large
throughout the ocean and varied less between seas. The highest deposition flux occurred in the YS (1110.8 $\mu g \cdot m^{-2} \cdot year^{-1}$) and
the lowest occurred in the NWP (566.4 $\mu g \cdot m^{-2} \cdot year^{-1}$). Despite the relatively lower deposition flux in the NWP, it still exerted
a noticeable impact on the NWP. In contrast, coarse-mode soluble Fe was mainly distributed in marginal seas, and the
depositional flux in the BS (186.1 $\mu g \cdot m^{-2} \cdot year^{-1}$) was ~14 times higher than that in the NWP (12.9 $\mu g \cdot m^{-2} \cdot year^{-1}$). Across the
ocean, soluble Fe deposition fluxes were greater in the fine mode than in the coarse mode, at 611.4 and 22.9 $\mu g \cdot m^{-2} \cdot year^{-1}$,
respectively. As illustrated in Fig.S7, fine-mode soluble Fe was primarily contributed by land anthropogenic sources, with a
relative contribution exceeding 94% across all marine regions. The contribution of ship sources to the deposition of fine-mode
soluble Fe was greater than that of dust sources, ranging from 3-6% in the Chinese marginal seas, and up to 19.2% in the ECS
during the summertime when ship activities are dynamic. Coarse-mode soluble Fe was strongly influenced by dust, with a
seasonal average contribution of 7.0% over the sea areas, which can reach 39.9% in April when dusty weather is prevalent.

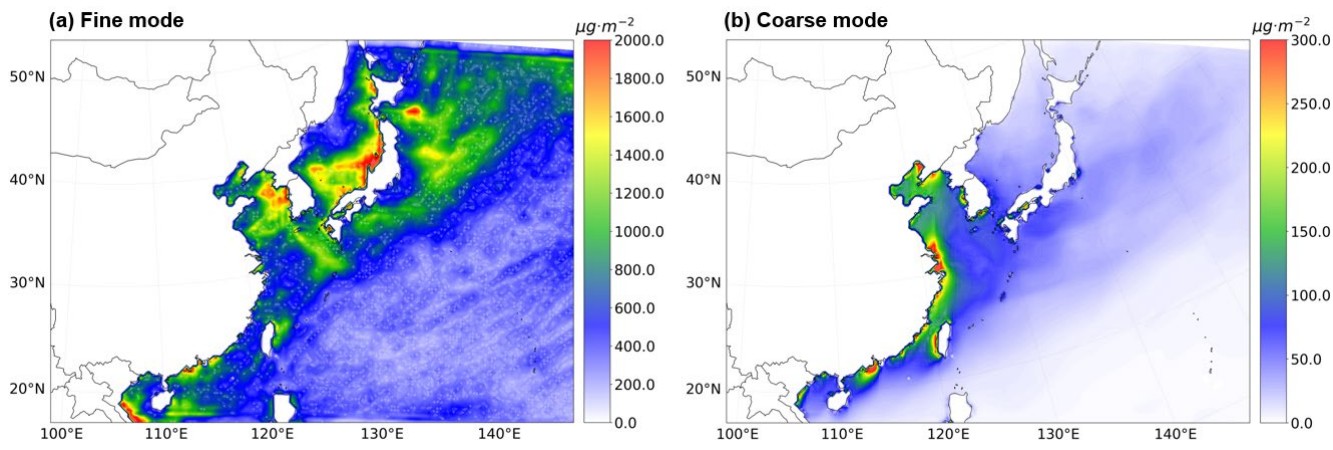


**Figure 5: Fine mode (a) and coarse mode (b) spatial distribution of the estimated soluble iron deposition fluxes throughout the year of 2017 (units: µg·m$^{-2}$·year$^{-1}$, including land anthropogenic, ship, and dust sources).**

On the one hand, aerosols emitted by anthropogenic sources are rich in acidic species such as NO$_x$ and sulfur dioxide (SO$_2$), whereas aerosols of dust tend to contain a significant portion of carbonates (Böke et al., 1999), which are much less acidic than anthropogenically sourced aerosols (Ito et al., 2019). For trace metals, acidity affects solubility through insoluble minerals readily dissolving under acidic conditions relevant to atmospheric aerosol (Baker et al., 2021; Hamilton et al., 2023; Li et al., 2017). On the other hand, smaller particles can undergo longer-distance transport in the atmosphere. Along with particle aging, metal morphology changes, and more metals dissolve. Besides, the emission of metals from anthropogenic sources was higher in the fine mode than coarse mode. The above reasons collectively lead to a higher deposition flux of soluble iron in the fine mode.

### 3.3 Sources and Sinks of Marine Metals

### 3.3.1 Budget of Trace Metals from Emission to Deposition

In Sect.3.1 and Sect.3.2, we discussed the contributions of the land anthropogenic, ship, and dust sources to the emissions, atmospheric concentrations, and deposition flux of six metal elements. This section focused on the source-sink patterns of metal elements in maritime areas. Figure 6 illustrated the proportional contributions of the three major sources to the entire area (land and ocean) emissions, atmospheric concentrations, and deposition of the six metals (percentages were calculated from a specific source divided by the total contribution of the three sources) in the sea areas and land areas, respectively.

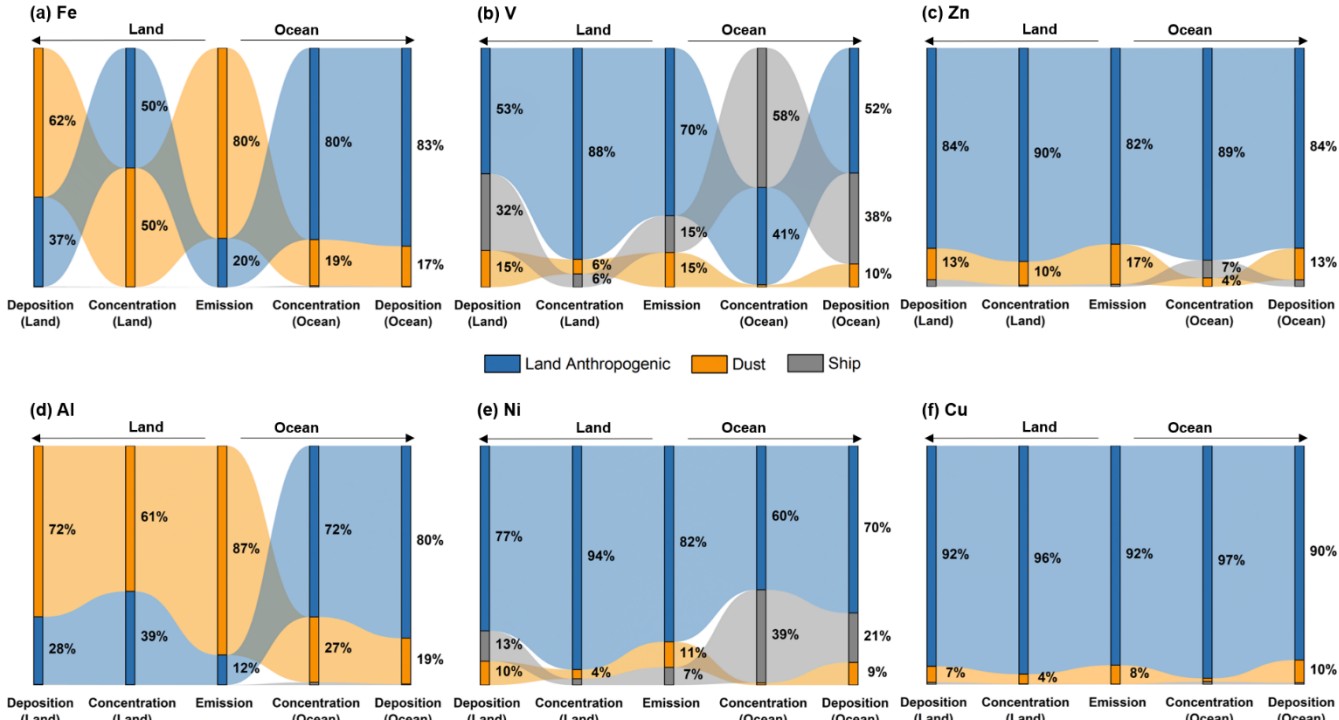

**Figure 6: Evolution of the relative contributions of the land anthropogenic, ship, and dust sources to emissions, seasonal mean**
**atmospheric concentrations, and annual deposition fluxes of Fe (a), V (b), Zn (c), Al (d), Ni (e), Cu (f) (Concentrations and**
**depositional fluxes labelled "Ocean" in the figure were for the oceans only, and concentrations and depositional fluxes labelled**
**"Land" were for land only).**
It can be found that for the predominant emission metals Fe and Al originating from dust sources, the contributions of dust
sources to emissions far exceeded that of land anthropogenic sources. However, as atmospheric transport processes occurred,
the contribution of land anthropogenic sources became significant and was comparable to the contribution of dust sources to
atmospheric concentrations. In particular, the contribution of land anthropogenic sources became dominant when focusing on
marine deposition. For Fe, the contribution from land anthropogenic sources was 20%, 80%, and 83% in the three stages from
emissions to marine deposition flux, similar to results reported by previous study (Kajino et al., 2020). Similarly, for Al, the
corresponding contributions were 12%, 72%, and 80%. The contributions from dust sources in marine deposition flux (17%
for Fe and 19% for Al) were much lower than those in emissions (80% for Fe and 87% for Al). Dust particles typically have
large particle sizes, making them more likely to deposit during atmospheric transport, which explains why, for all metals, the
contribution of dust sources in concentrations was lower than that in emissions over both the land and the ocean area. However,
because the dust source areas are mainly inland, such as Mongolia and northwestern China, the contribution of dust sources to
metal deposition in the sea was much less than that in the land area. To more accurately assess the impact of dust sources on
the budget of metals during the dust season (spring), we plotted the evolution of the same relative contributions for April
emissions, atmospheric concentrations, and deposition fluxes (Fig.S8). The contribution of dust sources to the spring marine

deposition fluxes of all metals became larger compared to the annual values, especially for Fe and Al, where the contribution exceeded 50%. This indicated that dust sources were the most important source of spring marine deposition fluxes for these two metals. However, the contribution of dust sources to metal deposition fluxes is significantly seasonal. On a year-round basis, dust sources were not the most important contributors to metal deposition fluxes in the East Asian Seas.

For metals such as V and Ni, the contributions from ship sources in marine deposition flux (38% and 21% respectively) were larger than those in emissions (15% and 7% respectively) and in deposition over the land area (32% and 13%, respectively). This reaffirmed the importance of ship sources when considering the metal deposition in the sea areas. Analyzing the contributions from the three sources revealed that despite the presence of dust source areas and high dust emissions in East Asia, the impact of dust on marine depositional fluxes was not as large as its impact on emissions. The contribution of land anthropogenic sources to maritime deposition flux was generally higher than that to emissions, except for V, where ship sources had a greater impact on deposition fluxes than on emissions. While it is true that dust sources contribute more metals, the impact of human activities on metal deposition is of greater concern when we focus on the East Asian seas.

### 3.3.2 Dominant Maritime Regions for the Three Major Emission Sources

The identification of the dominant sea area for sources was established based on the contributions of the three major sources to the marine deposition flux of metals. For each ocean grid in this study, the contribution rate of a source was calculated by dividing the metal deposition flux attributed to that source by the total deposition flux of the metal, thereby obtaining the contribution rate for the specific grid. The criteria were employed as follows. If one source contributed more than 66.7%, it was considered to dominate the metal deposition flux of the grid. If both sources contributed more than 33.3%, with the remaining one contributing less than 33.3%, it was considered that the two sources jointly dominated the deposition flux of the grid. And in the absence of dominance by one or two sources, it was considered that the three major sources collectively influenced the metal deposition flux of the grid.

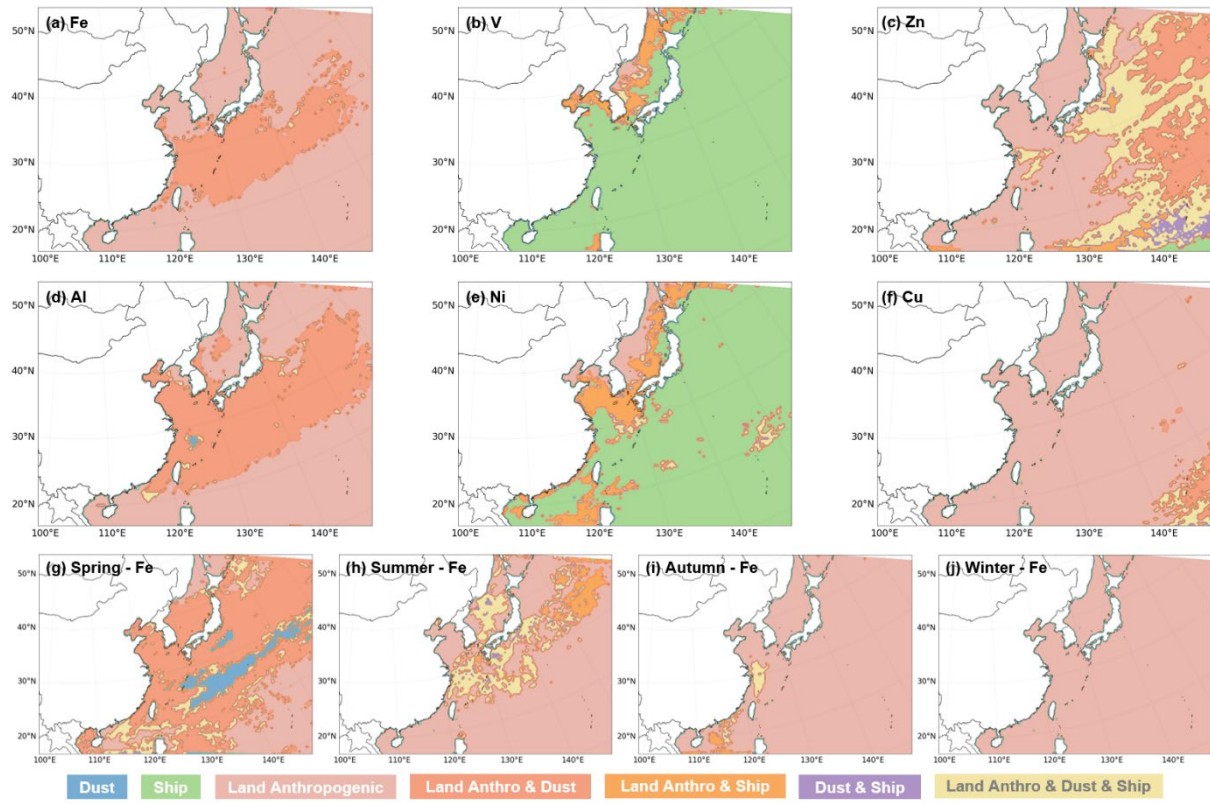

414

**Figure 7: The dominant source distributions of metal deposition fluxes in the ocean, Fe (a), V (b), Zn (c), Al (d), Ni (e), Cu (f); (g-j) are the dominant source of the deposition fluxes of soluble Fe in spring, summer, autumn and winter seasons. (In this study, we calculated the relative contributions of the metal sedimentation fluxes from the three major sources for each grid. A source was considered to dominate metal deposition on the grid if its contribution was > 67%, two sources were considered to jointly dominate metal deposition on the grid if both sources contributed > 33% and three sources were considered to jointly dominate in the rest of the cases).**

Based on the aforementioned calculation and criteria, the dominant sea areas for metal deposition fluxes from the three major sources were depicted in Fig.7. For Fe. Al, Zn, and Cu, land anthropogenic sources dominated the deposition fluxes in almost all offshore areas proximate to land. For V and Ni, a considerable range of metal deposition fluxes were dominated by both land anthropogenic and ship sources in offshore areas near land, especially in the BS, the YS, and the JS. In the vast majority of the open ocean area, the deposition of V and Ni was mainly dominated by ship sources. In contrast, for Fe and Al, there were scarcely any regions where land anthropogenic and ship sources co-dominate, but in the ECS and the NWP, a large range of metal deposition was co-dominated by both land anthropogenic and dust sources, similar to the previous result (Matsui et al., 2018). For Cu and Zn, the area dominated by land anthropogenic sources was extensive, especially for Cu, where land anthropogenic sources dominated the metal deposition fluxes in almost the entire ocean. Conversely, for Zn, areas still existed where both dust and land anthropogenic sources dominated, alongside areas where the three major sources collectively influenced the deposition fluxes in the western Pacific Ocean.

The main sources of six metallic elements were different, leading to the deposition of metallic elements in distinct oceanic
areas. Consequently, when assessing the ecological effects of a specific metal, it becomes particularly important to identify its
dominant emission sources. Mahowald et al. (2010) estimated that ocean primary productivity was enhanced by 6% due to the
doubling of desert dust which carried iron during the 20th century. When we combined this result on dust sources along with
our findings regarding the dominant source of soluble iron (see Figs.7g-7j) - the area of the East Asian Seas dominated by
anthropogenic sources of deposition was larger than that of dust sources - the resulting primary productivity of the East Asian
Seas may be more significant with the growing metal emissions from anthropogenic sources. Given that different metal
elements have distinct ecological effects in marine environments, it is crucial to consider their specific implications. For
example, the nutrient effect of Fe on marine primary productivity is a significant consideration (Bonnet et al., 2008; Mackey
et al., 2015; Mahowald et al., 2009; Schmidt et al., 2016; Yamamoto et al., 2022). For Cu, the focus may be on its toxicity or
synergistic effects with Fe on biophysiological processes (Guo et al., 2012; Wang et al., 2017a; Yang et al., 2019; Zou et al.,
2015). Zn, on the other hand, might be considered for its role in carbonic anhydrase and other biochemical processes (Morel
et al., 1994; Shaked et al., 2006; Tortell et al., 2000). Our identification of the main sources of metal deposition in sea waters
aids in investigating the potential ecological impacts.
**4 Conclusion**
Trace metals have a non-negligible impact on marine ecology, and their impact on marine productivity continues to be
explored. Due to the challenges of measuring atmospheric deposition fluxes in open seas, air quality models provide a solution
for this task. In this study, we established a monthly emission inventory covering six metal elements (Fe, Al, V, Ni, Zn, and
Cu) in the East Asian region (0-55°N, 85-150°E), incorporating land anthropogenic, ship, and dust sources. The CMAQ was
modified to assess the concentrations and deposition fluxes of metal species over the East Asian Seas and subsequently
estimated the soluble metal deposition fluxes, with a focus on the contributions of different sources across various sea regions.
We analyzed the evolutions in the relative contributions of the three sources to the six metals from source-emission to sink-
deposition and identified the dominant sources of deposition of the six metals in sea waters.
Throughout the year 2017, emissions from all sources were 1,021.5, 1,940.4, 11.7, 11.5, 27.2, and 14.0 kt of Fe, Al, V, Ni,
Zn, and Cu, respectively. The contribution of land anthropogenic sources to metal emissions was significant, exceeding 60%
for most metals, except for Fe and Al in the coarse mode, where the contributions from dust sources (80% and 87%,
respectively) were larger. Ship sources contributed more to V and Ni than to the remaining metals, mainly in the fine mode.
China was an emission hotspot for metallic elements within the modelled land area, and regions with dynamic ship activity
were emission hotspots for metals in the modelled sea area. The seasonal mean concentrations of Fe, Al, V, Ni, Zn, and Cu in
the sea areas were 34.87, 51.27, 0.95, 0.64, 0.98, and 0.49 ng·m$^{-3}$, respectively. And the concentrations of six metals over the
BS and the YS markedly surpassed those recorded in other seas, and were 6-60 times higher than those over the NWP. In
contrast, the deposition fluxes of the six metals varied much less over different sea areas, and can affect more remote waters,
such as the NWP. Pollutants carried by dust, especially Fe and Al, were transported to more open sea areas through intense
weather processes. The spatial distribution of deposition flux for these two metals in the sea areas was broader than that of the
remaining four metals. The estimated annual soluble deposition fluxes of Fe, Al, V, Ni, Zn, and Cu were 634.3, 1,701.6, 74.3,
46.1, 113.0, and 42.0 $\mu g \cdot m^{-2}$, respectively. The contribution of land anthropogenic sources to fine-mode soluble iron was
significant (> 94% across all sea areas), and dust sources contributed a lot to coarse-mode soluble iron (ranging from 18% to
74%). Particulate matter emitted by anthropogenic sources is more acidic than dust sources and is distributed in a higher
percentage in the fine mode, allowing for longer particle aging processes. As a result, higher soluble iron deposition fluxes in
the fine mode compared to the coarse mode.
Both land-based and marine-based anthropogenic sources (as known as shipping) played more important roles in maritime
deposition flux compared to emissions of trace metals. But the impact of dust on depositional fluxes was not as large as its
impact on emissions for East Asian seas. Land anthropogenic sources dominated or co-dominated the deposition of most
metals and soluble iron in East Asian seas. Ship sources dominated the deposition of V and Ni in most of the sea areas. Only
the soluble iron deposition in Spring was dust-dominated, which is associated with the seasonal characteristics of Asian dust,
mostly occurring in spring.
This study provides gridded data on atmospheric deposition fluxes with detailed source categories and identifies the
dominant source of metal deposition in the ocean for future assessments of the impact of trace metals on marine ecology. It
lays the foundation for a more profound understanding of the contributions of human activities and natural processes to metal
distribution in marine areas. Additionally, considering the different solubilities of metals from various sources, our source-
resolved data makes it possible to calculate soluble metal deposition flux on a source-by-source basis. However, further
research is still needed in the future to investigate the deposition, and solubility of metal elements in marine environments,
aiming to enhance the accuracy of estimates for soluble metal deposition flux.
**Author contribution**
Shenglan Jiang: Writing - original draft preparation, Investigation, Methodology, Software, Validation, Formal analysis, Data
curation, Visualization.
Yan Zhang: Conceptualization, Investigation, Supervision, Methodology, Validation, Formal analysis, Writing - review &
editing, Project administration, Funding acquisition.
Guangyuan Yu: Validation, Investigation, Writing review & editing.
Zimin Han: Data curation, Software.
Junri Zhao: Data curation, Investigation, Methodology.
Tianle Zhang: Data curation, Writing - review & editing.
Mei Zheng: Writing – review, Funding acquisition.

**Code/Data availability**

The Final Analysis (FNL) meteorological data from are available from National Centres for Environmental Predictions (NCEP) at https://rda.ucar.edu/datasets/ds083.2. The base source code of CMAQv5.4 is available at https://github.com/USEPA/CMAQ. The model data presented in this paper can be obtained from Yan Zhang (yan_zhang@fudan.edu.cn) upon request.

**Competing interests**

The authors declare that they have no conflict of interest.

**Acknowledgments**

The work was supported by the National Natural Science Foundation of China (No. 42375100, No. 42030708), the Natural Science Foundation of Shanghai Committee of Science and Technology, China (No. 22ZR1407700), and the Program of Pudong Committee of Science and Technology, Shanghai (No. PKJ2022-C05).

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
