# Peer review of "Source-resolved atmospheric metal emissions, concentrations, and their deposition fluxes into the East Asian Seas"

_EGUsphere, 2024_

## Author Comment (AC1)

**Response to Reviewer 2 Comments**

Dear Reviewer,

We would like to express our sincere gratitude for your time to review our manuscript and for the helpful suggestions to improve the article entitled "***Source-resolved atmospheric metal emissions, concentrations, and their deposition fluxes into the East Asian Seas***". We have carefully reviewed all the comments and revised the article accordingly. Please find the detailed responses below in blue and the corresponding revisions in track changes in the revised manuscript.

The reviewer's comments are in black.

The author's responses are in blue.

Revisions in the manuscript are in ***italics and bold*** (line numbers before and inside the bracket refer to those in revised manuscript with and without track of changes, respectively).

**Specific comments:**

1.  First, the validation of the modeling results should be further discussed, which I think is very important for modeling work. Tables S9-10 were not discussed sufficiently. If there are measurements on surface concentrations of metals over lands and oceans, some detailed comparisons are needed (site/region/total average). You can refer to some similar works on N emissions, in which the validation works were made between modeling and measurements on land and oceans (see the supporting materials of papers: https://doi.org/10.1073/pnas.2121998119; https://doi.org/10.1073/pnas.2221459120).

    **Response:**

    Thank you for pointing this out. As you suggested, we have added a comparative analysis of the long-term monitoring data of $PM_{2.5}$-bound metals from the Shanghai Pudong site with the atmospheric metal concentrations obtained from the modelling to Page 10, Lines 254-258 of the revised manuscript, with a comparative graph Figure S4 (Supplementary Information Page 6, Lines 50-52). In addition to concentration comparisons over land, we used navigational measurements of atmospheric iron concentrations from the sea area for validation, as there are

few long-term measurements in the sea area. On Page 10, Lines 252-254 of the revised manuscript *"The concentration of Fe was 201.1 ng·m⁻³ in the YS and 92.17 ng·m⁻³ in the ECS, and the contribution of land anthropogenic sources to the Fe concentration was 71.6% in the ECS, similar with the values reported by previous study (Zhang et al., 2024)."*

**Revisions in the manuscript:**

**1. Page 11 (10), Lines 273-276 (254-258):**

*The available long-term and near real-time concentration monitoring data of V and Ni in the fine mode at the Pudong site (in Shanghai, China) obtained by Zou et al. (2020) were used to further validate the simulation of the model. As presented in Fig.S4, the simulated concentrations of V and Ni were in good agreement with the monitoring data, with respective normalized mean fractional bias (NMFB) and normalized mean fractional error (NMFE) of -0.31 and 0.37 for V and -0.38 and 0.40 for Ni.*

**2. Supplementary Information Page 6, Lines 50-52 (50-52):**

*Figure S4. Comparison of simulated daily concentrations of V (a) and Ni (b) with observations at the Pudong site (31.2331°N, 121.5447°E, Shanghai, China).*

[Figure]

2. Second, are there any social-economic drivers for seasonal changes in metal emissions? For instance, when we talked about $NH_3$, it's usually controlled by increasing population and food production (N fertilizer, livestock). $NH_3$ seasonal changes are mainly affected by temperature and fertilizer applications. I hope to see some additional discussions on metal seasonal changes. How the urbanization affects metal emissions and pollution? (see refs on social-economic drivers on agricultural N emissions: L. Liu. 2023 Nature, https://doi.org/10.1038/d41586-023-02753-9; Deng et al. 2024 Nature communications, https://doi.org/10.1038/s41467-023-44685-y).

**Response:**

Thanks for your suggestions. After reading the literature you recommended we can understand the effect of socio-economic drivers on the seasonal characteristics of emissions. Especially for $NH_3$, which is highly correlated with agriculture, factors such as urbanization, population, fertilizer application, etc. can significantly affect emissions. As the base year of the emission inventory established in this study is 2017, there is no long-term data to support the analysis of the impact of industrialization progress or urbanization on emissions. Therefore, we limit our discussion to the seasonal changes of emissions, and a detailed discussion has been added on Page 6, Lines 166-183 of the revised manuscript.

**Revisions in the manuscript:**

**1. Pages 6-7 (6), Lines 179-197 (166-183):**

[revised manuscript text omitted]

---

## Author Comment (AC2)

**Response to Reviewer 1 Comments**

Dear Reviewer,

We would like to express our sincere gratitude for your time to review our manuscript and for the thoughtful comments and helpful suggestions to improve the article entitled "***Source-resolved atmospheric metal emissions, concentrations, and their deposition fluxes into the East Asian Seas***". We have carefully reviewed all the comments and revised the article accordingly. Please find the detailed responses below in blue and the corresponding revisions in track changes in the revised manuscript.

The reviewer's comments are in black.

The author's responses are in blue.

Revisions in the manuscript are in ***italics and bold*** (line numbers before and inside the bracket refer to those in revised manuscript with and without track of changes, respectively).

**Major concern:**

Development of emission inventory (Sections 2.2 and 3.1): I can follow the methodology itself; however, we should recognize that dust emission can largely varied year-to-year. From the simulation, only four months (January, April, July, and October) were conducted, but annual emissions were presented in Fig. 1. How did the authors estimate annual total emissions? Or, the covering spring time is enough to calculate the annual total emissions as stated? Moreover, this study targeted the year of 2017, but why? The estimated dust emissions were possible maxima/minima or average situation for Asia dust? These details explanations are required to follow this study.

**Response:**

Thanks for your questions. The first question is about how to estimate annual total emissions. We categorized the sources of metal emissions into land anthropogenic, ship, and dust sources, where emissions from land anthropogenic and ship sources were calculated for each month of 2017. For dust emissions, our original calculation method used the simulation output data for January, April, July, and October to represent the monthly emissions for the corresponding seasons to obtain the

annual dust emissions.

Following your suggestions and in consideration of the seasonal variations in dust emissions, we have supplemented monthly simulation experiments to calculate the dust emissions for the entire year. The revised methodology for calculating metal emissions in 2017 has been added on Page 4, Lines 130-131 and Page 5, Lines 154-156 of the revised manuscript (Revisions 1-4), and supplementary calculated monthly dust metal emissions Table S8-9 have been added on Pages 15-16, Lines 88-91 of the Supplementary Materials. The updated annual average $PM_{10}$ dust emissions of 13.49 μg/m$^2$/s is similar to that calculated by Zhao et al. (2013). Furthermore, discussion of seasonal variations of dust emissions has been added on Page 6, Lines 175-183 of the revised manuscript (Revisions 5-7). In addition, we have updated the corresponding data in Figures 1a and 1b (Page 7, Line 184), Figure 2 (Page 9, Line 222) and Figure 6 (Page 17, Line 374), as well as the related discussion, based on the recalculated annual dust emissions (Revisions 8-14). To more accurately assess the impact of dust sources on metal emissions, atmospheric concentrations, and deposition fluxes, we have also added spring-specific discussions in Sections 3.1.1, 3.2, and 3.3.1 in the revised manuscript. The relevant revisions can be found in the responses to Comments 9, 10, and 11.

The second question is about the targeted year of this study. We chose 2017 for our modelling experiments, mainly because of the year in which the anthropogenic emission inventory data were available. For land anthropogenic sources, we used the EDGAR and SPECIATE databases. Although both databases are regularly updated, they still do not cover recent years. The latest available annual and monthly sector-specific gridmaps provided by EDGAR are for 2018 (https://edgar.jrc.ec.europa.eu, accessed on 1 April 2024), while the most recent data available at the time this study was conducted was for 2017. And from 1 January 2018, the Domestic Emission Control Area (DECA) policy began to be phased in, requiring ships to use low-sulfur fuel to reduce emissions. In 2020, the International Maritime Organization (IMO) low-sulfur fuel regulations were implemented globally. Therefore, our calculations of ship emissions for 2017 are more representative of ship emissions over a long period until 2018.

**Revisions in the manuscript:**

**1. Page 5 (4), Lines 125-128 (118-121):**

*The general methodology for calculating **monthly** land anthropogenic emissions of metals was to*

*multiply each source of PM emissions by the fraction of the metal content in PM. **Monthly emissions data for 2017 for** each source category **of PM** was provided by the Emissions Database for Global Atmospheric Research (EDGAR) emission inventories, ...*

**2. Page 5 (4), Lines 132-134 (125-126):**

***The monthly emission inventory of metals from*** *ship sources was established by a bottom-up approach based on **real-time data from the Automatic Identification of Ships (AIS) database for the year 2017** (Yuan et al., 2023; Zhao et al., 2020).*

**3. Page 5 (4), Lines 138-139 (130-131):**

***The monthly*** *dust emissions of trace metals **in 2017** were generated from in-line modules developed by Foroutan et al. (2017) during the CMAQ run.*

**4. Page 6 (5), Lines 163-167 (154-156):**

***We used monthly emission inventories from land anthropogenic and ship sources and modelled monthly dust emissions for 2017 to calculate metal emissions for the entire year. The relative contribution of the three sources to metal emissions and the seasonal variation characteristics were assessed, and then emissions from land anthropogenic sources were further specified.***

**5. Supplementary Information Pages 16-17 (15-16), Lines 91-94 (88-91):**

*Table S8. Monthly fine mode metal emissions from dust sources in 2017 (Unit: tons·month$^{-1}$)*

| Month | Cu | Fe | V | Ni | Zn | Al |
|-------|-------|----------|-------|-------|---------|----------|
| Jan | 3.65 | 680.75 | 2.43 | 0.41 | 26.38 | 1155.55 |
| Feb | 21.75 | 4054.75 | 14.50 | 2.42 | 157.11 | 6882.81 |
| Mar | 2.27 | 423.72 | 1.52 | 0.25 | 16.42 | 719.25 |
| Apr | 142.17 | 26499.65 | 94.78 | 15.80 | 1026.81 | 44982.26 |
| May | 36.11 | 6730.91 | 24.07 | 4.01 | 260.81 | 11425.49 |
| Jun | 5.84 | 1088.05 | 3.89 | 0.65 | 42.16 | 1846.92 |
| July | 43.86 | 8174.67 | 29.24 | 4.87 | 316.75 | 13876.23 |
| Aug | 10.20 | 1901.40 | 6.80 | 1.13 | 73.68 | 3227.55 |
| Sep | 18.53 | 3453.22 | 12.35 | 2.06 | 133.81 | 5861.73 |
| Oct | 15.09 | 2811.89 | 10.06 | 1.68 | 108.96 | 4773.09 |
| Nov | 0.88 | 164.37 | 0.59 | 0.10 | 6.37 | 279.00 |

| | | | | | |
|---|---|---|---|---|---|
| *Dec* | *15.00* | *2796.45* | *10.00* | *1.67* | *108.36* | *4746.88* |
| *Sum* | *315.36* | *58779.82* | *210.24* | *35.04* | *2277.61* | *99776.76* |

*Table S9. Monthly coarse mode metal emissions from dust sources in 2017 (Unit: tons·month⁻¹)*

| Month | Cu | Fe | V | Ni | Zn | Al |
|---|---|---|---|---|---|---|
| *Jan* | *9.17* | *8734.20* | *17.25* | *13.75* | *26.69* | *18492.78* |
| *Feb* | *54.59* | *52023.79* | *102.76* | *81.89* | *158.96* | *110149.13* |
| *Mar* | *5.70* | *5436.48* | *10.74* | *8.56* | *16.61* | *11510.56* |
| *Apr* | *356.79* | *339998.86* | *671.60* | *535.18* | *1038.89* | *719874.11* |
| *May* | *90.62* | *86359.67* | *170.59* | *135.94* | *263.88* | *182847.93* |
| *Jun* | *14.65* | *13959.97* | *27.58* | *21.97* | *42.66* | *29557.21* |
| *July* | *110.06* | *104883.59* | *207.18* | *165.09* | *320.48* | *222068.34* |
| *Aug* | *25.60* | *24395.50* | *48.19* | *38.40* | *74.54* | *51652.20* |
| *Sep* | *46.49* | *44305.93* | *87.52* | *69.74* | *135.38* | *93808.24* |
| *Oct* | *37.86* | *36077.48* | *71.26* | *56.79* | *110.24* | *76386.26* |
| *Nov* | *2.21* | *2108.86* | *4.17* | *3.32* | *6.44* | *4465.05* |
| *Dec* | *37.65* | *35879.35* | *70.87* | *56.48* | *109.63* | *75966.76* |
| *Sum* | *791.41* | *754163.67* | *1489.71* | *1187.11* | *2304.39* | *1596778.58* |

**6. Supplementary Information Page 4, Lines 41-43 (41-43):**

[revised manuscript text omitted]

**Specific comments:**

1. Line 30: The component of trace metals should be first defined in the first appearance (not in the second sentence).

**Response:**

Thank you for pointing this out. We have adjusted the order of the words, defining the component of trace metals in the first sentence (Page 1, Line 30), and changed the wording of the second sentence from "*Trace metals (iron, cobalt, nickel, copper, zinc, manganese, cadmium, lead, and rare earth elements, among others) are present in seawater at very low concentrations, ...*" to "*They are present in seawater at very low concentrations, ...*" (Page 1, Lines 30-31).

**Revisions in the manuscript:**

**1. Page 2 (1), Lines 33-34 (30-31):**

*Trace metals **(iron, cobalt, nickel, copper, zinc, manganese, cadmium, lead, and rare earth elements, among others)** have been the focus of marine biogeochemical studies for half a century. **They** are present in seawater at very low concentrations, ....*

2. Line 49: A high temperature of aerosol itself? Please clarify.

**Response:**

Sorry for the misunderstanding. The intention was to convey that PM-bound metals emitted from anthropogenic sources are typically emitted through combustion sources before being released. The term 'high temperature' has been clarified as it is ambiguous and may cause confusion.

**Revisions in the manuscript:**

**1. Page 2, Lines 52-54 (48-50):**

*By contrast, aerosols emitted from anthropogenic sources **are often produced by high-temperature combustion and** are characterized by  small particle sizes (Bowie et al., 2009; Chen et al., 2012; Li et al., 2017; Oakes et al., 2012).*

3. Line 58-63: But this study was conducted over East Asia. This paragraph seems to be mainly focused on the Southern Ocean. The motivation for East Asia is also required to understand the introduction of this study.

**Response:**

Thanks for your suggestion. We have adjusted Paragraph 3 of the Introduction on Pages 2-3, Lines 58-68 of the revised manuscript to include the motivation for selecting the East Asian Seas as the study area.

**Revisions in the manuscript:**

**1. Pages 2-3, Lines 62-73 (58-68):**

*The spatial distribution of metal emissions from ship and anthropogenic sources, contrasts with that of dust (Mahowald et al., 2018).* ***Dust has long been considered an important source of Fe to the surface ocean, particularly in remote areas away from continental margins (Jickells et al., 2005). However,*** *Matsui et al. (2018) suggested that anthropogenic Fe may dominate the total deposition flux of soluble Fe and its variability over southern oceans* ***(30-90°S) by incorporating recent measurements of anthropogenic magnetite into a global aerosol model, which increased the estimated total deposition flux of soluble Fe to southern oceans by 52%. Pinedo-González et al. (2020) determined from iron-stable isotopes that anthropogenic Fe contributed 21-59% of soluble Fe measured in the North Pacific Ocean. The Northwest Pacific is located directly downwind of the industrially active East Asian region with significant and increasing metal emissions and is influenced by westerly winds transporting Asian dust (often mixed with anthropogenic aerosol and gases) (Hamilton et al., 2023). Identifying the dominant sources of metal deposition in the ocean is important for estimating soluble metal deposition, especially in the East Asian seas with significant contributions from both dust and anthropogenic metal emissions.***

4. Line 78 or Line 84: The relevant information (e.g., doi of zenodo) for the original CMAQ modeling system itself is needed here.

**Response:**

Thanks for pointing this out and the relevant information has been included in the revised manuscript (Page 3, Lines 89-90).

**Revisions in the manuscript:**

**1. Page 4 (3), Lines 96-97 (89-90):**

*The CMAQ* ***(E.P.A, 2020)*** *is a widely used air quality model that encompasses a wide range of*

*complex atmospheric physicochemical processes.*

5. Line 84-90: Because this study analyzed deposition, a description of the deposition scheme in the CMAQ should be presented.

**Response:**

Thank you for the suggestions to give a more comprehensive description of the CMAQ configuration. We have added a detailed description of the deposition scheme on Pages 3-4, Lines 98-100 of the revised manuscript.

**Revisions in the manuscript:**

**1. Page 4 (3-4), Lines 105-107 (98-100):**

*M3Dry scheme was used to calculate dry deposition (Pleim and Ran, 2011), and the aerosol dry deposition model was upgraded in version 5.4, showing better comparison with size-resolved observations (Pleim et al., 2022); AQCHEM cloud chemistry was used to calculate wet deposition (Fahey et al., 2017).*

6. Line 121: I guess the inline dust module in the CMAQ for Foroutan et al. (2017) (doi:10.1002/2016MS000823) is required, or did the authors develop their models?

**Response:**

Thanks for pointing this out. We used the inline dust module developed by Foroutan et al. (2017), to which citations have been added to the revised manuscript (Page 4, Lines 130-131).

**Revisions in the manuscript:**

**1. Page 5 (4), Lines 138-139 (130-131):**

*The monthly dust emissions of trace metals in 2017 were generated from in-line modules developed by Foroutan et al. (2017) during the CMAQ run.*

7. Line 131-133: I do not fully understand this sentence.

**Response:**

Sorry that sentence was a bit ambiguous, we have tweaked the wording to make it clearer (Page 5, Lines 138-142). The sentence expresses that Kurisu et al. (2021) collected samples of total and soluble iron and used the stable iron isotope approach to identify the contribution of dust

and anthropogenic sources, respectively, in order to calculate the solubility of iron emitted from the two sources separately.

**Revisions in the manuscript:**

**1. Page 5, Lines 146-152 (138-142):**

*Kurisu et al. (2021) **used the stable Fe isotope source apportionment method to analyze dust Fe and anthropogenic Fe concentrations in total and soluble Fe samples. The results** showed that the **solubility of** dust Fe in the Northwest Pacific Ocean ranged from 0.9 ~ 1.3% (dust-contributed soluble Fe divided by dust-contributed total Fe) and 11% for **solubility of** anthropogenic Fe (anthropogenic-contributed soluble Fe divided by anthropogenic-contributed total Fe).*

8. Line 185: Before starting this section 3.1.2, it would be better to mention Fig. 2 at first (not in Line 201).

**Response:**

As you suggested, we have mentioned Fig.2 at the beginning of Sect 3.1.2 (Page 8, Line 210).

**Revisions in the manuscript:**

**1. Page 9 (8), Line 225 (210):**

***The spatial distributions of metal emissions were presented in Figs.2a-2f.*** *For the entire simulation area, ...*

9. Line 215-218: In addition to the targeted seas such as ECS explained here, further discussion focusing on the springtime would clarify the importance of dust emissions.

**Response:**

Thank you for the helpful suggestions. We do think that an analysis of the spring dust contributions could highlight the importance of dust sources, which would be more valid than an analysis of the relative contributions of dust sources to the ECS. Therefore, we have added Figure S3 in the Supplementary Information on Page 5, Lines 45-48, and have added further discussion of the contribution of springtime dust sources to atmospheric metal concentrations on Page 10, Lines 244-248, and Page 11, Lines 273-274 of the revised manuscript, respectively.

**Revisions in the manuscript:**

**1. Page 11 (10), Lines 261-265 (244-248):**

*Asian dust storms occur annually in late winter and spring in the main dust regions of the Gobi Desert, Taklamakan Desert, and Loess Plateau (Hsu et al., 2010). Therefore, dust sources played a more significant role in April, contributing 39.2% of the Fe and 51.3% of the Al concentrations in the sea area covered by the study. In the East China Sea (ECS), these values could reach 48.3% and 67.8%, respectively (as presented in Fig.S3).*

**2. Supplementary Information Page 5, Lines 45-48 (45-48):**

*Figure S3. Absolute and relative contributions of seasonal mean concentrations of Fe (a) and Al (b) in different sea areas from land anthropogenic, ship, and dust sources (units: ng·m⁻³), the numbers on top of the stacked bar graphs represent total seasonal mean concentrations from three sources.*

[Figure]

**3. Page 12 (11), Lines 292-293 (273-274):**

*As shown in Fig.S3, dust sources contributed 40.8% and 50.3% of the atmospheric concentrations of Fe and Al in the NWP in spring, respectively.*

10. Line 228-230 (the caption of Figure 3) and the relevant discussion: I do not follow why this estimation is expressed as the total annual mean concentration. Because this study was only conducted for four months, even though these are representative months of each season, the wording "annual mean" will be overstated. In addition to this question, what is the meaning of "total"? If this is the total concentration, the concentration for the "ALL" region should be the

sum of all ocean areas. Please clarify these expressions and the actual analyzed contents.

**Response:**

Thanks for your questions. As you say, these four months are representative of the corresponding seasons and have been used in previous studies to calculate "annual mean concentrations" (Cai et al., 2021; Lane and Pandis, 2007; Li and Xie, 2016; Lv et al., 2018; Wang et al., 2015; Wen et al., 2016; Xie et al., 2008), so this method has been used in our study as well. However, we agree that the term "annual mean" is a little bit overstated, so we have amended it to "seasonal mean" in the caption of Figure 3 (Page 11, Lines 262-264) and the corresponding text (Page 17, Lines 375-376 and Page 20, Lines 460-461) for greater accuracy. And we have added a note on Page 9, Lines 235-238 of the revised manuscript that this estimate leads to a slight overestimation of the dust source contribution.

In response to your second question, we have explained "total concentration" in the caption of Figure 3, which represents "*the total seasonal mean concentrations from the three major sources*". The term "Total" represents the sum of atmospheric metal concentrations contributed by land anthropogenic, dust, and ship sources.

**Revisions in the manuscript:**

**1. Page 11 (9), Lines 252-255 (235-238):**

*Overall, the **seasonal mean** metallic concentrations in sea areas were 34.9, 51.3, 1.0, 0.6, 1.0, and 0.5 ng·m$^{-3}$ for Fe, Al, V, Ni, Zn, and Cu, respectively. **It is worth noting that we chose January, April, July, and October to represent each of the four seasons, and since most of the spring dust events in East Asia occur in April, this estimate would result in a slight overestimation of the contribution of dust sources.***

**2. Page 12 (11), Lines 280-282 (262-264):**

*Figure 4: Contributions of **seasonal** mean concentrations of metallic elements in different sea areas from land anthropogenic, ship, and dust sources, Fe (a), V (b), Zn (c), Al (d), Ni (e), Cu (f) (units: ng·m$^{-3}$), with the numbers at the top of the stacked bar charts representing the total **seasonal** mean concentrations from the three major sources.*

**3. Page 19 (17), Lines 401-404 (375-376):**

*Figure 5: Evolution of the relative contributions of the land anthropogenic, ship, and dust sources to emissions, **seasonal** mean atmospheric concentrations, and annual deposition fluxes*

*of Fe (a), V (b), Zn (c), Al (d), Ni (e), Cu (f).*

**4. Page 23 (20), Lines 493-494 (460-461):**

*The **seasonal** mean concentrations of Fe, Al, V, Ni, Zn, and Cu in the sea areas were 34.87, 51.27, 0.95, 0.64, 0.98, and 0.49 ng·m$^{-3}$, respectively.*

**11.** Line 257-260 (the caption of Figure 4) and the relevant discussion: Same comment to the above comment on Line 228-230.

**Response:**

Thanks for pointing this out. We used the simulation output data for January, April, July, and October to represent the monthly deposition flux for the corresponding seasons to obtain the annual deposition flux, and this estimation method has been used in previous studies (Lin et al., 2010; Zhang et al., 2010). Given the considerable seasonal variability of dust sources, we revised the methodology for estimating annual deposition fluxes from dust sources based on supplementary calculations of annual dust emissions. When using monthly deposition fluxes to estimate seasonal values, we employed a multiplier between the total seasonal emissions from dust sources and the emissions from a representative month as a conversion factor. The method and references for estimating annual deposition fluxes have been added on Page 12, Lines 286-293 of the revised manuscript. Based on the revised estimation methodology, we have correspondingly revised all the figures involving annual deposition fluxes as well as the discussion (Revisions 2-9). To more accurately assess the impact of dust sources on metal deposition fluxes, we have added spring-specific discussions in Sections 3.2.2 and 3.3.1 in the revised manuscript (Revisions 10-12). In order to clarify that the annual deposition fluxes in this study were derived from estimates, we have specified "annual deposition flux" as "estimated annual deposition flux" on Page 13, Line 298, and Page 14, Line 321 of the revised manuscript.

**Revisions in the manuscript:**

**1. Page 13 (12), Lines 306-313 (286-293):**

[revised manuscript text omitted]

**11. Supplementary Information Page 11 (10), Lines71-75 (68-72):**

**Figure S8. Evolution of the relative contributions of the land anthropogenic, ship, and dust sources to emissions, atmospheric concentrations, and deposition fluxes of Fe (a), V (b), Zn (c), Al (d), Ni (e), Cu (f) for the month of April (Concentrations and depositional fluxes labelled "Ocean" in the figure were for the oceans only, and concentrations and depositional fluxes labelled "Land" were for land only).**

[Figure]

**12. Page 20 (17-18), Lines 417-424 (390-396):**

*To more accurately assess the impact of dust sources on the budget of metals during the dust season (spring), we plotted the evolution of the same relative contributions for April emissions, atmospheric concentrations and deposition fluxes (Fig. S8). The contribution of dust sources to the spring marine deposition fluxes of all metals became larger compared to the annual values, especially for Fe and Al, where the contribution exceeded 50%. This indicated that dust sources were the most important source of spring marine deposition fluxes for these two metals. However, the contribution of dust sources to metal deposition fluxes is significantly seasonal. On a year-round basis, dust sources were not the most important contributors to metal deposition fluxes in the East Asian Seas.*

**12.** Line 288-313: Again, why these estimations can be explained as annual amounts? Taking into consideration the important role of Asian dust in spring, how about the additional analyses for soluble Fe deposition flux focusing on springtime?

**Response:**

Thank you for the question. In our response to Comment 11, we explained the reasonableness of using deposition fluxes from representative months of the four seasons to estimate annual deposition fluxes and revised the methodology for estimating annual deposition fluxes from dust sources. In the revised manuscript, we used the revised estimated annual deposition fluxes from Section 3.2.2 to calculate soluble metal deposition fluxes, and accordingly, the soluble

iron deposition fluxes contributed by dust sources were also revised. All figures and discussions involving soluble deposition fluxes have been modified in the revised manuscript (Revisions 1-5 and 7-8).

In addition to this, as you suggested, we have added analyses of the contribution of dust sources to soluble Fe deposition fluxes in the spring on Page 15, Lines 351-355 of the revised manuscript (Revision 6).

**Revisions in the manuscript:**

**1. Page 17 (15), Lines 361-363 (340-342):**

*Table 1: Marine deposition fluxes of soluble metals in fine and coarse particulate forms (Units: $\mu g \cdot m^{-2} \cdot year^{-1}$)*

|  | *Cu* | *Fe\** | *Zn* | *V* | *Ni* | *Al* |
|---|---|---|---|---|---|---|
| *Fine* | *26.1* | *611.4* | *80.2* | *72.4* | *41.8* | *1608.9* |
| *Coarse* | *15.8* | *22.9* | *32.7* | *1.9* | *4.3* | *92.7* |

*\*The soluble iron deposition flux was calculated separately for each of the three sources and then summed to obtain the total soluble deposition flux*

**2. Page 16 (14-15), Lines 356-360 (335-338):**

*Land anthropogenic, ship, and dust sources contributed 600.0, 10.6, and **1.7** $\mu g \cdot m^{-2} \cdot year^{-1}$ of soluble Fe in the fine mode and 10.9, 0, and **12.0** $\mu g \cdot m^{-2} \cdot year^{-1}$ of soluble Fe in the coarse mode, respectively. Based on this method, the solubility of Fe (soluble Fe from all three sources divided by total Fe deposition flux) obtained in this study ranged from **4% to 17%**, ...*

**3. Page 17 (15), Lines 367-372 (346-351):**

*The highest deposition flux occurred in the YS (**1110.8** $\mu g \cdot m^{-2} \cdot year^{-1}$) and the lowest occurred in the NWP (**566.4** $\mu g \cdot m^{-2} \cdot year^{-1}$). Despite the relatively lower deposition flux in the NWP, it still exerted a noticeable impact on the NWP. In contrast, coarse-mode soluble Fe was mainly distributed in marginal seas, and the depositional flux in the BS (**186.1** $\mu g \cdot m^{-2} \cdot year^{-1}$) was ~**14** times higher than that in the NWP (**12.9** $\mu g \cdot m^{-2} \cdot year^{-1}$). Across the ocean, soluble Fe deposition fluxes were greater in the fine mode than in the coarse mode, at **611.4 and 22.9** $\mu g \cdot m^{-2} \cdot year^{-1}$, respectively.*

**4. Page 18 (15-16), Lines 380-383 (356-358):**

[Figure]

*Figure 8: Fine mode (a) and coarse mode (b) spatial distribution of **the estimated** soluble iron deposition fluxes throughout the year of 2017 (units: μg·m⁻²·year⁻¹, including land anthropogenic, ship, and dust sources).*

**5. Supplementary Information Page 9, Lines 64-67 (63-66):**

*Figure S7. Absolute and relative contributions of soluble iron deposition fluxes from land anthropogenic, ship, and dust sources in different sea areas, fine mode (a), coarse mode (b) (units: μg·m⁻²·year⁻¹), the numbers on top of the stacked bar graphs represent total deposition fluxes from three sources.*

[Figure]

**6. Page 17 (15), Lines 372-379 (351-355):**

***As illustrated in Fig.S7,*** *fine-mode soluble Fe was primarily contributed by land anthropogenic*

*sources, with a relative contribution exceeding 94% across all marine regions. **The contribution of ship sources to the deposition of fine-mode soluble Fe was greater than that of dust sources, ranging from 3-6% in the Chinese marginal seas, and up to 19.2% in the ECS during the summertime when ship activities are dynamic.** Coarse-mode soluble Fe was strongly influenced by dust, with **a seasonal average contribution of 52.3% over the sea areas, which can reach 39.9% in April when dusty weather is prevalent.***

**7. Page 24 (21), Lines 499-502 (466-469):**

*The estimated annual soluble deposition fluxes of Fe, Al, V, Ni, Zn, and Cu were **634.3, 1,701.6, 74.3, 46.1, 113.0, and 42.0 μg·m⁻²**, respectively. The contribution of land anthropogenic sources to fine-mode soluble iron was significant (> 94% across all sea areas), and dust sources contributed a lot to coarse-mode soluble iron (ranging from **18% to 74%**).*

**8. Page 1, Line 21 (21):**

*…, soluble deposition fluxes were **634.3, 1,701.6, 74.3, 46.1, 113.0, and 42.0 μg·m⁻²**, respectively.*

**13.** Line 289-292: I do not follow where the targeted seas to this estimation. Please specify.

**Response:**

Thank you for pointing this out, it is necessary to clarify the targeted sea area for estimating soluble metal deposition fluxes. We have specified this on Page 14, Lines 333-334 of the revised manuscript.

**Revisions in the manuscript:**

**1. Page 16 (14), Lines 354-355 (333-334):**

*Utilizing the calculation methods in Sect.2.3, the detailed **results** of these calculations **for soluble metal deposition fluxes to the ocean within the study area** were provided in Table 1.*

**14.** Line 293: What is the meaning of "final", and how to evaluate again the solubility?

**Response:**

Thanks for your question. The methodology for calculating soluble Fe deposition fluxes is described in Sect. 2.3 of the manuscript, where we multiply the Fe deposition fluxes contributed by land anthropogenic, dust, and ship sources by the solubility of Fe from the respective sources, respectively. However, marine input of Fe is not contributed by a single source, so the solubility

of total Fe contributed by all sources needs to be calculated, which is the "final solubility of Fe"
in the original manuscript. That is the sum of the calculated soluble Fe deposition fluxes from
the three sources divided by the total Fe deposition fluxes from the three sources. The word
"final" is indeed ambiguous and we have added an explanation about calculating the solubility
of Fe on Pages 14-15, Lines 337-338 of the revised manuscript.

**Revisions in the manuscript:**

**1. Page 16 (14-15), Lines 358-360 (337-338):**

*Based on this method, the  solubility of Fe **(soluble Fe from all three sources divided by**
**total Fe deposition flux)** obtained in this study ranged from 4% to 17%*

15. Line 323: It might be better to reconsider this subsection title.

    **Response:**

    As you suggested, we have reconsidered the title of this subsection (Page 16, Line 368).

    **Revisions in the manuscript:**

    **1. Page 19 (16), Line 393 (368):**

    *3.3.1 Budget of Trace Metals from Emission to Deposition*

16. Line 332-343: This kind of analysis is interesting, but it is still hard to understand the result. I
    am a little bit confused because the deposition over land should be considered in Figure 6. So,
    how about to show the concentration over land and ocean, and the deposition over land and
    ocean separately (not as "All")?

    **Response:**

    Thanks for your question, it makes us realize that this figure is not clear enough. As you
    suggested, we have presented both atmospheric concentrations of metals as well as deposition
    fluxes separately over land and ocean (Pages 16-17, Lines 371-378 Figure 6). We have adjusted
    the corresponding analyses in the revised manuscript, specifically in the second and third
    paragraphs of Section 3.1.1 (Page 17, Lines 383-386, and Lines 388-390, and Page 18, Lines
    381-383).

    **Revisions in the manuscript:**

    **1. Page 19 (16-17), Lines 396-404 (371-378):**

*Figure 6 illustrated the proportional contributions of the three major sources to the entire area (land and ocean) emissions, **marine atmospheric** concentrations, **and deposition of** the six metals (percentages were calculated from a specific source divided by the total contribution of the three sources) **in the sea areas and land areas, respectively.***

[Figure]

*Figure 9: Evolution of the relative contributions of the land anthropogenic, ship, and dust sources to emissions, **seasonal** mean atmospheric concentrations, and annual deposition fluxes of Fe (a), V (b), Zn (c), Al (d), Ni (e), Cu (f) **(Concentrations and depositional fluxes labelled "Ocean" in the figure were for the oceans only, and concentrations and depositional fluxes labelled "Land" were for land only).***

**2. Pages 19-20 (17), Lines 409-413 (383-386):**

*For Fe, the contribution from land anthropogenic sources was **20%, 80%, and 83%** in the **three** stages from emissions to marine deposition flux, similar to results reported by previous study (Kajino et al., 2020). Similarly, for Al, the corresponding contributions were **12%, 72%, and 80%**. The contributions from dust sources in **marine** deposition flux (17% for Fe and **19%** for Al) were much lower than those in emissions (**80%** for Fe and **87%** for Al).*

**3. Page 20 (17), Lines 415-417 (388-390):**

*However, because the dust source areas are mainly inland, such as Mongolia and northwestern China, the contribution of dust sources to metal deposition in the sea was much less than that in the **land** area.*

**4. Page 20 (18), Lines 425-427 (397-399):**

*For metals such as V and Ni, the contributions from ship sources in **marine** deposition flux (38%*
*and 21% respectively) were larger than those in emissions (15% and 7% respectively) and in*
*deposition over the **land** area (32% and 13%, respectively).*

**17.** Line 425: "emissions" of what?

**Response:**

Thanks for the correction. We have added the missing information on Page 21, Lines 473 of the
revised manuscript.

**Revisions in the manuscript:**

**1. Page 24 (21), Lines 505-506 (472-473):**

*Both land-based and marine-based anthropogenic sources (as known as shipping) played more*
*important roles in maritime deposition flux compared to emissions **of trace metals**.*

**18.** Line 432-434: The final remark was ambiguous. How to enhance the accuracy of soluble metal
deposition flux? What is the contribution of this study to the future study?

**Response:**

Thanks for your question. We have added a further explanation on Page 21, Lines 481-482 to
make it clear. It is more accurate to use the contribution from different sources multiplied by
the solubility of that source separately than to use the total deposition flux directly to calculate
the soluble metal deposition flux. Our study provides data on source-resolved seasonal metal
deposition fluxes, which offers the possibility of refined calculations of soluble metal deposition.

**Revisions in the manuscript:**

**1. Page 24 (21), Lines 515-516 (481-482):**

[revised manuscript text omitted]

Li, W., Xu, L., Liu, X., Zhang, J., Lin, Y., Yao, X., Gao, H., Zhang, D., Chen, J., Wang, W., Harrison, R. M., Zhang, X., Shao, L., Fu, P., Nenes, A., and Shi, Z.: Air pollution–aerosol interactions produce more bioavailable iron for ocean ecosystems, Science Advances, 3, e1601749,

10.1126/sciadv.1601749, 2017.

Lin, C. J., Pan, L., Streets, D. G., Shetty, S. K., Jang, C., Feng, X., Chu, H. W., and Ho, T. C.: Estimating mercury emission outflow from East Asia using CMAQ-Hg, Atmos. Chem. Phys., 10, 1853-1864, 10.5194/acp-10-1853-2010, 2010.

Lv, Z., Liu, H., Ying, Q., Fu, M., Meng, Z., Wang, Y., Wei, W., Gong, H., and He, K.: Impacts of shipping emissions on PM2.5 pollution in China, Atmos. Chem. Phys., 18, 15811-15824, 10.5194/acp-18-15811-2018, 2018.

Mahowald, N. M., Hamilton, D. S., Mackey, K. R. M., Moore, J. K., Baker, A. R., Scanza, R. A., and Zhang, Y.: Aerosol trace metal leaching and impacts on marine microorganisms, Nature Communications, 9, 2614, 10.1038/s41467-018-04970-7, 2018.

Matsui, H., Mahowald, N. M., Moteki, N., Hamilton, D. S., Ohata, S., Yoshida, A., Koike, M., Scanza, R. A., and Flanner, M. G.: Anthropogenic combustion iron as a complex climate forcer, Nature Communications, 9, 1593, 10.1038/s41467-018-03997-0, 2018.

Oakes, M., Ingall, E. D., Lai, B., Shafer, M. M., Hays, M. D., Liu, Z. G., Russell, A. G., and Weber, R. J.: Iron Solubility Related to Particle Sulfur Content in Source Emission and Ambient Fine Particles, Environmental Science & Technology, 46, 6637-6644, 10.1021/es300701c, 2012.

Pinedo-González, P., Hawco, N. J., Bundy, R. M., Armbrust, E. V., Follows, M. J., Cael, B. B., White, A. E., Ferrón, S., Karl, D. M., and John, S. G.: Anthropogenic Asian aerosols provide Fe to the North Pacific Ocean, Proceedings of the National Academy of Sciences, 117, 27862-27868, 10.1073/pnas.2010315117, 2020.

Pleim, J. and Ran, L.: Surface Flux Modeling for Air Quality Applications, Atmosphere, 2, 271-302, 10.3390/atmos2030271, 2011.

Pleim, J. E., Ran, L., Saylor, R. D., Willison, J., and Binkowski, F. S.: A New Aerosol Dry Deposition Model for Air Quality and Climate Modeling, Journal of Advances in Modeling Earth Systems, 14, e2022MS003050, doi.org/10.1029/2022MS003050, 2022.

Wang, P., Cao, J., Tie, X., Wang, G., Li, G., Hu, T., Wu, Y., Xu, Y., Xu, G., Zhao, Y., Ding, W., Liu, H., Huang, R., and Zhan, C.: Impact of Meteorological Parameters and Gaseous Pollutants on PM2.5 and PM10 Mass Concentrations during 2010 in Xi'an, China, Aerosol and Air Quality Research, 15, 1844-1854, 10.4209/aaqr.2015.05.0380, 2015.

Wen, W., Cheng, S., Liu, L., Chen, X., Wang, X., Wang, G., and Li, S.: PM2.5 Chemical

Composition Analysis in Different Functional Subdivisions in Tangshan, China, Aerosol and Air Quality Research, 16, 1651-1664, 10.4209/aaqr.2015.09.0559, 2016.

Xie, S. D., Liu, Z., Chen, T., and Hua, L.: Spatiotemporal variations of ambient $PM_{10}$ source contributions in Beijing in 2004 using positive matrix factorization, Atmos. Chem. Phys., 8, 2701-2716, 10.5194/acp-8-2701-2008, 2008.

Yuan, Y., Zhang, Y., Mao, J., Yu, G., Xu, K., Zhao, J., Qian, H., Wu, L., Yang, X., Chen, Y., and Ma, W.: Diverse changes in shipping emissions around the Western Pacific ports under the coeffect of the epidemic and fuel oil policy, Science of The Total Environment, 879, 162892, 10.1016/j.scitotenv.2023.162892, 2023.

Zhang, Y., Yu, Q., Ma, W., and Chen, L.: Atmospheric deposition of inorganic nitrogen to the eastern China seas and its implications to marine biogeochemistry, Journal of Geophysical Research: Atmospheres, 115, doi.org/10.1029/2009JD012814, 2010.

Zhang, Y., Mahowald, N., Scanza, R. A., Journet, E., Desboeufs, K., Albani, S., Kok, J. F., Zhuang, G., Chen, Y., Cohen, D. D., Paytan, A., Patey, M. D., Achterberg, E. P., Engelbrecht, J. P., and Fomba, K. W.: Modeling the global emission, transport and deposition of trace elements associated with mineral dust, Biogeosciences, 12, 5771-5792, 10.5194/bg-12-5771-2015, 2015.

Zhao, C., Chen, S., Leung, L. R., Qian, Y., Kok, J. F., Zaveri, R. A., and Huang, J.: Uncertainty in modeling dust mass balance and radiative forcing from size parameterization, Atmos. Chem. Phys., 13, 10733-10753, 10.5194/acp-13-10733-2013, 2013.

Zhao, J., Zhang, Y., Xu, H., Tao, S., Wang, R., Yu, Q., Chen, Y., Zou, Z., and Ma, W.: Trace Elements From Ocean-Going Vessels in East Asia: Vanadium and Nickel Emissions and Their Impacts on Air Quality, Journal of Geophysical Research: Atmospheres, 126, e2020JD033984, 10.1029/2020JD033984, 2021.

Zhao, J., Zhang, Y., Patton, A. P., Ma, W., Kan, H., Wu, L., Fung, F., Wang, S., Ding, D., and Walker, K.: Projection of ship emissions and their impact on air quality in 2030 in Yangtze River delta, China, Environmental Pollution, 263, 114643, 10.1016/j.envpol.2020.114643, 2020.